

# Dynamic token encryption for preventing permission leakage in serverless architectures

Yu Liu[1], Fu Li[1,2] and Chenhao Sun[3]

[1] School of Computer Science, National University of Defense Technology, Changsha, Hunan, China
[2] Lab, Intelligent Game and Decision Lab (IGDL), Tianjin, China
[3] Alibaba Cloud, Alibaba, Hangzhou, Zhejiang, China

## ABSTRACT

Serverless architecture simplifies application development and operation, but its permission control model based on static execution roles struggles to adapt to highly dynamic runtime environments, which can easily lead to the risk of permission and key leakage. To address this challenge, this article proposes a runtime dynamic token-based access control scheme. The scheme combines function context and user-defined security rules to achieve function-level dynamic authorization and request-level identity authentication. The generated dynamic tokens possess strong randomness, unpredictability, and one-time use characteristics, effectively reducing the harm caused by token leakage. Moreover, the designed multi-factor token verification model integrates dynamic factors such as call chain features and behavior patterns, which can defend against various security threats. Through social surveys, qualitative analysis, and extensive experiments, this article confirms that the proposed scheme significantly enhances the security of serverless applications while maintaining a controllable impact on platform performance. This research enriches the theoretical knowledge in the field of serverless security and provides new ideas for development practices, which is expected to promote the expansion of serverless architecture to enterprise-level scenarios and contribute to the healthy development of its ecosystem.

## INTRODUCTION

In recent years, serverless computing has gained widespread attention in both academia and industry due to its unique advantages, such as automated resource management, on-demand scaling, and reduced operational overhead (*Eismann et al., 2020*; *Kritikos & Skrzypek, 2018*; *Li, Leng & Chen, 2022*). An increasing number of enterprises are choosing to migrate their applications to serverless platforms to lower operational costs and improve resource utilization. However, the serverless computing model also faces unique security challenges, particularly in the aspect of permission management (*de Oliveira, 2022*; *Calles, 2020*; *Li, Leng & Chen, 2022*).

Corresponding author
Fu Li, lifu@xmail.tech

Permission management is the cornerstone of security protection in serverless platforms. The platform needs to configure corresponding execution permissions for each function to support flexible access to required cloud services. However, the complex permission configuration process not only increases the burden on development and operation but also easily introduces security vulnerabilities (*Barrak et al., 2024*). Improper configuration can lead to functions obtaining excessive permissions and even being exploited by attackers to access sensitive data, posing serious threats to user privacy and data security. Moreover, traditional static access control mechanisms struggle to effectively adapt to the highly dynamic and uncertain nature of serverless applications (*Sankaran, Datta & Bates, 2020*). In practice, developers are often forced to grant higher permissions to functions to avoid impacting business functionality, which, however, lays the groundwork for permission abuse.

Permission management defects bring severe security risks to serverless platforms and applications. Our survey shows that 27% of serverless users have experienced various types of attacks, 55% of users are concerned about the risk of permission leakage, and 48% of users express concern about the practice of storing execution permission keys through environment variables. The industry has witnessed sensitive data leakage incidents caused by improper permission management in serverless environments, where attackers exploited overly permissive execution roles to gain unauthorized access to cloud resources (*Barrak et al., 2024*). These incidents highlight the inadequacies of existing mechanisms in terms of dynamicity, flexibility, and defense-in-depth.

Academia has conducted a series of explorations on serverless security issues, focusing on aspects such as execution environment isolation, cold start security, and access control optimization (*Datta et al., 2020*; *Alpernas et al., 2018*). These approaches have made important contributions to improving serverless security. However, existing work is either limited to specific scenarios or struggles to adapt to the dynamicity of serverless platforms, and cannot fundamentally solve the core problem of permission management (*Jegan et al., 2020*). For example, information flow control approaches (*Alpernas et al., 2018*) provide strong security guarantees but face challenges in practical implementation across different serverless platforms. Similarly, workflow-based security solutions (*Datta et al., 2020*; *Sankaran, Datta & Bates, 2020*) offer valuable protection but may not fully address the dynamic nature of function-level permissions.

In the industry, mainstream serverless platforms provide some security enhancement mechanisms, but their static and predefined policies are difficult to fully adapt to the flexibility and variability of serverless applications. The complex permission configuration and narrow usage scenarios also greatly reduce the usability and universality of these mechanisms. Traditional methods such as JSON Web Tokens (JWT) and key management services (KMS) are not specifically designed for the unique characteristics of serverless architectures, particularly the ephemeral nature of function instances and their dynamic execution contexts.

In summary, how to achieve efficient, fine-grained, and dynamic serverless permission management without sacrificing function flexibility is a critical challenge that needs to be addressed urgently. This is also the main research motivation of this article. To tackle this

challenge, this article proposes an innovative dynamic token scheme, aiming to achieve further control over permissions by introducing dynamically generated temporary tokens at runtime. Unlike static authorization mechanisms, our approach incorporates function execution context and request-specific information to generate unique tokens for each operation, significantly enhancing security while maintaining flexibility.

The scheme combines function context information and custom rules to generate highly dynamic and unpredictable tokens and introduces multi-factor authentication in the request verification phase to build a defense-in-depth access control system. This approach differs from traditional token-based mechanisms by tightly coupling tokens with both function execution context and request parameters, making them inherently resistant to replay attacks and unauthorized access attempts.

The main contributions of this article are as follows:

(1) We propose a novel access control scheme based on runtime dynamic tokens that addresses the permission management challenges in serverless platforms, providing enhanced granularity and dynamicity compared to existing static permission models;

(2) We develop a token generation mechanism that integrates function context information and user-defined rules, creating tokens with high dynamicity, time-effectiveness, and unpredictability, which effectively resist common attacks such as replay attacks;

(3) We introduce a multi-factor verification process that combines request context analysis and behavioral patterns to create a comprehensive defense-in-depth permission control system;

(4) We implement and evaluate a prototype system on mainstream serverless platforms, demonstrating through comprehensive experiments that our approach effectively enhances security while maintaining acceptable performance overhead.

The rest of this article is organized as follows: "Serverless Computing Model and Security Challenges" introduces the basic concepts and security challenges of serverless computing; "Implementation Principles of Dynamic Tokens" reviews existing serverless security protection schemes and their limitations; "Security and Performance Evaluation" elaborates on the system architecture and key technologies of the proposed dynamic token scheme; "Discussion" comprehensively evaluates the security and performance of the scheme around typical attack scenarios; "Conclusion" summarizes the work and outlines future research directions.

# SERVERLESS COMPUTING MODEL AND SECURITY CHALLENGES

## Security threat analysis of serverless platforms

The unique system architecture and working methods of serverless platforms expose them to a series of new security threats that traditional security protection mechanisms struggle to effectively address (*de Oliveira, 2022*; *Calles, 2020*; *Li, Leng & Chen, 2022*). These threats

mainly stem from the inherent characteristics of serverless platforms in terms of permission management, user code execution, and data flow.

First, the permission management mechanism of serverless platforms has inherent flaws that can lead to multiple security risks. Serverless platforms need to configure corresponding execution permissions for each function to support flexible access to required cloud services, such as object storage and databases. The working logic of this execution permission is mainly that when a function instance is created and started, the serverless platform generates temporary keys based on the user-configured execution permissions and permission system and configures the temporary keys into environment variables for direct use by the user. However, the highly dynamic and unpredictable nature of serverless applications means that overly fine-grained permission configuration often increases the learning and usage costs for users (*Barrak et al., 2024*). In reality, developers are often forced to configure relatively lax permission sets for functions to avoid impacting business functionality, even if the developer tools provided officially often default to configuring relatively lax permissions for user experience, which lays the groundwork for permission abuse and leakage. According to the OWASP Serverless Top 10 (*OWASP Foundation, 2018*), broken access control is a critical vulnerability where "granting functions access to unnecessary resources or excessive permissions on resources is a potential backdoor to the system." Excessively broad permission configurations significantly increase the potential attack surface, allowing attackers to exploit leaked high-privilege keys to illegally access other users' sensitive data and critical resources, posing serious threats to user privacy and data security (*Alpernas et al., 2018*).

In the industry, permission leakage incidents on serverless platforms have been frequent, and major security incidents caused by permission management vulnerabilities have occurred in the actual operation of some serverless platforms (*Datta et al., 2020*). For example, a well-known cloud service provider's serverless deployment tool supports the synchronous deployment of multiple serverless functions and surrounding related services. To enhance user experience, the originally locally-executed tool was changed to execute in the cloud. Since the tool is a component-based tool that needs to load user-declared components during execution to complete corresponding capabilities, it allows users to customize deployment components through configuration files. As a result, attackers can construct malicious components to steal temporary keys in environment variables of the deployment service during the deployment process. Using these keys, attackers can illegally access code and configuration information uploaded by other users on the account corresponding to the service. Another mainstream cloud service provider's WebIDE platform is implemented based on serverless services. Originally, relevant execution permissions were required during function startup to perform some initialization operations, and the execution permissions would be automatically cleared after completion. However, after an upgrade of the serverless service provider's platform, due to an incremental change in key naming conventions that caused compatibility issues, the temporary key cleanup logic of the WebIDE platform failed. Attackers can thus obtain platform-level temporary keys in the WebIDE and gain execution permissions for the

function, illegally reading other users' function logs, which often contain business logic and sensitive data. Although cloud service providers took remedial measures after the incidents occurred, these incidents reflect the inherent vulnerability of serverless platforms in key management. The impact of such incidents often extends to the security of the entire cloud account (*Jegan et al., 2020*). Once core systems or databases are compromised, the business continuity and data confidentiality of users will be seriously threatened, which is undoubtedly disastrous for enterprises that heavily rely on cloud services.

To summarize, user code on serverless platforms often originates from complex sources, making it difficult to ensure there are no potential vulnerabilities. The OWASP Serverless Top 10 (*OWASP Foundation, 2018*) highlights that "attackers will try to look for a forgotten resource, like a public cloud storage, or open APIs" and that "secrets could be accidentally uploaded to the github repo, put it on a public bucket or even used hardcoded in the function." Attackers can trigger function execution through carefully crafted malicious events and induce function code containing vulnerabilities to leak high-privilege key information, thereby gaining access to cloud platform resources. Moreover, the event-driven data flow method of serverless platforms also introduces new security risks (*Alpernas et al., 2018*). Sensitive data frequently flows between distributed functions. If end-to-end encryption protection and integrity verification are lacking, it can easily be hijacked and tampered with by intermediaries.

Furthermore, log auditing and anomaly detection on serverless platforms face numerous challenges. According to the OWASP Serverless Top 10 (*OWASP Foundation, 2018*), "applications which do not implement a proper auditing mechanism and rely solely on their service provider probably have insufficient means of security monitoring and auditing." Traditional security information and event management systems struggle to effectively adapt to the high dynamicity and short instance lifecycle of serverless platforms. Current serverless platforms still have obvious shortcomings in providing fine-grained, comprehensive log recording and intelligent security analysis, making it very difficult to restore and investigate security incidents.

In summary, the unique technical characteristics and working methods of serverless platforms expose them to many new security threats that traditional security solutions struggle to effectively cover. The OWASP Serverless Top 10 provides a framework for understanding these threats, covering aspects such as injection, broken authentication, sensitive data exposure, broken access control, security misconfiguration, and insufficient logging and monitoring (*Li, Leng & Chen, 2022*). These threats involve multiple aspects such as permission management, code execution, data flow, and log auditing, and they intertwine to form a complex security threat landscape. There is an urgent need to explore a more flexible, dynamic, and fine-grained new paradigm of security protection that specifically addresses the disconnect between static permission models and highly dynamic execution environments. This paradigm must incorporate runtime context into access decisions, implement request-level permission validation, and provide defense-in-depth through multi-factor verification. This is also the starting point and focus of this article's work.

## Existing solutions and their limitations

To address the security issues in serverless architecture, academia and industry have conducted explorations at multiple levels. Following a systematic classification approach (*Li, Leng & Chen, 2022*), these efforts can be organized into platform-level protection, function-level controls, and application-level security enhancements.

One line of work focuses on security hardening of the serverless platform itself, such as permission management at the scheduling level (*Li et al., 2022*), security isolation mechanisms (*Zhang et al., 2019*), and timely cleanup of instance space (*Alzayat et al., 2023*). These efforts aim to eliminate potential security risks from the underlying architecture by improving the platform's own security protection capabilities. Another line of work focuses on peripheral monitoring, alerting, and auditing mechanisms, such as security event notification (*Agache et al., 2020*) and behavior anomaly detection, aiming to discover and respond to potential security issues in a timely manner by real-time monitoring of the running state of serverless applications.

In terms of fine-grained access control, existing work has actively explored the issues of improper permission management and insufficient access control that are common in serverless platforms. Excessive permission settings and coarse-grained authorization models are the main reasons for potential unauthorized access and sensitive information leakage in serverless applications (*Govind & González–Vélez, 2021*). To this end, some work proposes using fine-grained security policies and dynamic authorization mechanisms to strengthen permission control in serverless platforms. Notable approaches include Valve (*Datta et al., 2020*), which enables fine-grained control of information flows in function workflows through network-layer monitoring, and will.iam (*Sankaran, Datta & Bates, 2020*), which implements a workflow-aware access control model at the point of ingress. However, traditional centralized authorization services are often tightly coupled with application code, which may affect the overall performance of the system (*Sabbioni et al., 2022*). Although decentralized authorization solutions can alleviate performance bottlenecks to a certain extent, they may introduce new issues such as management complexity and cross-environment compatibility when actually implemented.

Furthermore, some research attempts to enhance the security of serverless applications from the application architecture level. For example, literature (*Ouyang et al., 2023*) proposes combining the microservice architecture with serverless computing. By splitting applications into multiple fine-grained function services and utilizing the short-lived instances and resource isolation mechanisms of the serverless platform, the security of cross-domain transactions can be enhanced. However, the introduction of this architecture may further increase system complexity, and the technical requirements for development and operation personnel are also higher.

Some inherent technical characteristics of serverless computing may also introduce new security risks. For example, the automatic elastic scaling mechanism common in serverless platforms significantly improves resource utilization, but if effective monitoring and protection are lacking, it may be exploited by malicious users to launch targeted Distributed Denial of Service (DDoS) attacks (*Wang et al., 2022*). Additionally, to solve the

problem of sensitive data sharing and protection in serverless applications, some studies propose introducing information flow control (IFC) technology (*Alpernas et al., 2018*). IFC achieves fine-grained security protection of data flow by security labeling data objects and computing entities and enforcing access control based on predefined flow rules. However, the actual deployment of IFC requires deep integration with the access control mechanisms of various cloud services. How to implement a unified and compatible IFC labeling system across different serverless platforms is an urgent problem to be solved.

When examining conventional security technologies in the context of serverless characteristics, fundamental limitations emerge. Serverless computing features event-driven execution, ephemeral instances, high-frequency cold starts, dynamic scaling, and multi-tenant environments. These characteristics create unique security challenges that traditional mechanisms struggle to address. For instance, JWT (JSON Web Tokens) typically operate on a time-based validity model, where tokens remain valid for predetermined periods (minutes to hours). This approach fundamentally misaligns with serverless functions' millisecond-level execution duration, creating a significant security vulnerability window where leaked tokens remain valid long after the function execution completes. In multi-tenant serverless environments, this time-scale mismatch significantly amplifies the risk of token misuse.

Similarly, key management services (KMS) face inherent limitations when applied to serverless architectures. The per-invocation KMS API calls introduce latency that can be disproportionate to the actual function execution time, potentially becoming a performance bottleneck during rapid scaling events. This is particularly problematic in serverless's pay-per-use economic model, where security-related overhead directly impacts operational costs. Furthermore, when applications are decomposed into dozens of fine-grained functions, configuring and managing KMS access policies for each function becomes exceedingly complex, often leading to overly permissive settings that compromise security.

Existing solutions exhibit several critical limitations when viewed through the lens of serverless computing's unique characteristics: (1) temporal mismatch between security mechanisms designed for long-running services and the millisecond-scale execution of serverless functions; (2) inability to efficiently handle the high-frequency state changes inherent in serverless environments; (3) security overhead disproportionate to function execution time; (4) coarse-grained permission models inadequate for microfunction architectures; and (5) lack of cost-effective validation mechanisms compatible with the pay-per-use model.

A particularly significant limitation of traditional approaches is their reliance on time-based rather than invocation-based security controls. Time-based tokens like JWT remain valid for extended periods, creating unnecessary exposure windows, while serverless functions typically complete execution in milliseconds. In contrast, an invocation-based approach where tokens are generated per request and immediately invalidated after use would align more naturally with serverless execution patterns and significantly reduce the potential impact of token leakage.

Future research needs to explore a new paradigm of cloud-native security protection that is flexible, scalable, and cross-platform consistent based on the unique technical characteristics of serverless itself (*Li, Leng & Chen, 2022*). Such a paradigm should address the fundamental disconnect between existing security models and the serverless execution environment by incorporating execution context into security decisions, implementing fine-grained request-level authorization, and providing strong protection against credential theft and misuse, all while maintaining the performance benefits that make serverless computing attractive. This is also the focus of this article's work. Only by developing new security mechanisms that are compatible with serverless from the kernel architecture and working mechanisms can we truly achieve the trusted development of serverless computing.

# IMPLEMENTATION PRINCIPLES OF DYNAMIC TOKENS

## Design philosophy

### Dynamic token generation mechanism

The dynamic token encryption and decryption mechanism consists of two main components: the generation of encrypted tokens and the verification of decrypted tokens. The generation of encrypted tokens is based on user-defined encryption rules, while the verification of decrypted tokens depends not only on user-defined encryption rules but also on additional information such as random strings and timestamps. The entire process involves dynamic and unpredictable authentication information exchanges, which is why this method is called the dynamic token encryption and decryption mechanism.

The core of the encrypted token generation mechanism is to provide a flexible and secure way to generate encrypted tokens, aiming to enhance the security of function invocations in serverless architecture. This mechanism allows for user-defined rules, including the integration of built-in dynamic parameters and custom static parameters, to achieve accurate verification and release of sensitive operations, further ensuring the security of serverless applications.

Specifically, as shown in Fig. 1, the token generation process allows users to dynamically generate signature strings by utilizing built-in parameters (such as request ID, instance ID, and timestamp) in the serverless environment combined with custom rules. For example, users can use the expression $md5($requestid, $instanceid) to set the generation rule for the signature string. This is merely one example of how dynamic parameters can be combined; users can define various rules based on their security requirements and choose different cryptographic algorithms according to their needs. The rule will exhibit different encrypted token values in different instances of the same function, and even under different requests of the same function instance.

Figure 1's architecture diagram illustrates the complete function lifecycle and permission management process. This process can be divided into three main phases:

First is the function creation and configuration phase (pink section, step 1): Users create functions and provide metadata and code, while defining encryption rules in the configuration. The function metadata, including encryption rule configuration, is stored in the platform for subsequent processing.

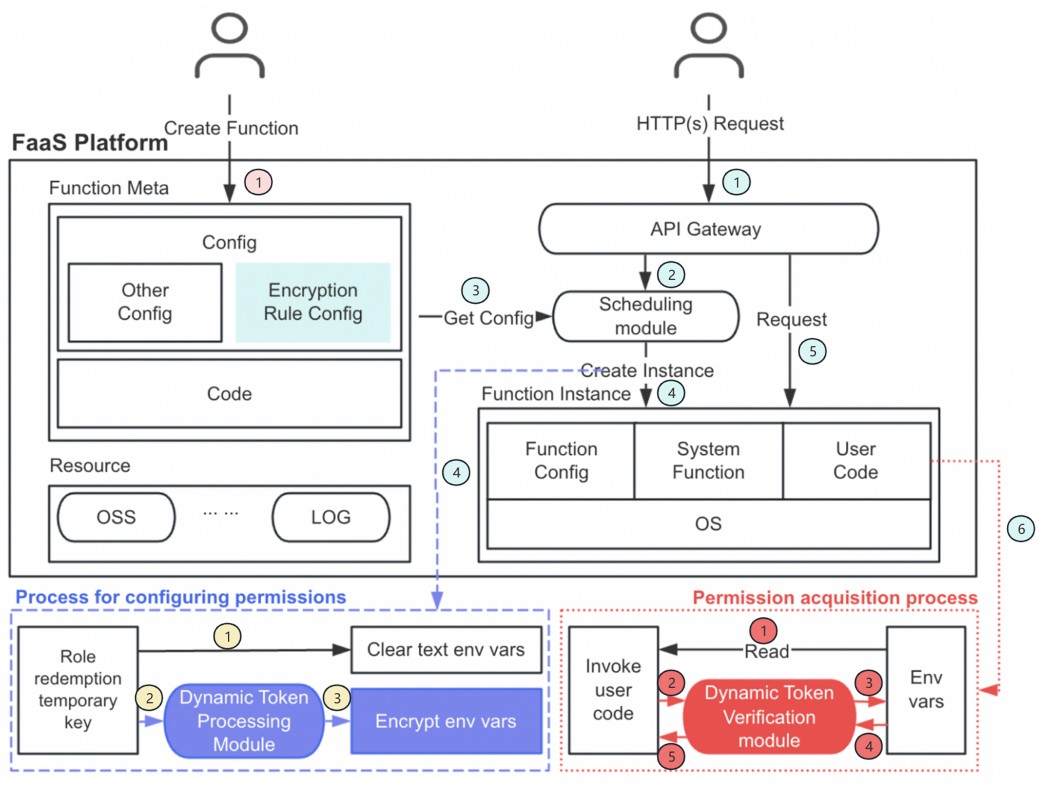

**Figure 1 Dynamic token encryption architecture diagram.**

Next is the cold start process (blue section, steps 1–4), which includes the permission configuration phase (blue dashed box): When an HTTP(s) request arrives (step 1), the API gateway forwards it to the scheduling module (step 2). The scheduling module retrieves the configuration information (step 3) and creates the corresponding function instance (step 4). During instance creation, the system performs role redemption to obtain temporary keys and immediately clears plaintext environment variables to eliminate direct exposure risks (yellow section, step 1). Subsequently, the dynamic token processing module processes these temporary keys according to predefined encryption rules (yellow section, steps 2–3), generating encrypted environment variables that are securely stored. This process ensures that sensitive information is protected from the very beginning, rather than existing in plaintext form in the function environment.

Finally, there is the permission acquisition phase (red dotted box, steps 1–4) that occurs after the cold start completes (step 5 in the blue section initiates normal function invocation, leading to step 6 for sensitive information acquisition): When the user code is invoked and needs to access protected resources (red section, step 1), it attempts to read environment variables. The dynamic token verification module intercepts this request (red section, step 2), comprehensively validates the provided verification information (red section, step 3), and only after all security checks pass does it return the decrypted environment variable values (red section, step 4), allowing the code to securely access the required resources.

When requesting sensitive information, users need to provide the decryption token, a non-repeatable random string within the instance, a timestamp, and the key values of the required parameters. More formally, if we denote $H$ as the cryptographic hash function chosen by the user (such as MD5, SHA-256, or any other algorithm based on security requirements), and | as the concatenation operator, the decryption token can be represented as:

$$DecryptToken = H(RequestID|InstanceID|Nonce|Timestamp|DataKey). \tag{1}$$

This formulation clearly illustrates how multiple dynamic factors are combined to create a unique, context-bound token. It's important to note that this is a flexible framework rather than a fixed implementation. The parameters included in the equation represent a recommended combination, but users can adjust the specific components based on their security needs—adding additional contextual factors or simplifying the combination for performance considerations. Similarly, the concatenation operation shown here can be replaced with more complex combining methods if desired. The essential requirement is consistency between token generation and verification processes.

The components of this equation are defined in Table 1.

The system generates a verification key through the same formula using the submitted parameters and matches it against the provided decryption token. To prevent replay attacks, the system designs an anti-replay mechanism that verifies the uniqueness of the random string (Nonce) and confirms the timestamp is within a reasonable validity period. This anti-replay implementation is straightforward, typically maintaining a simple list of used random strings within the function instance process for the duration of the instance's lifecycle.

In the dynamic token mechanism, the management of random numbers (Nonce) and timestamps is key to building replay protection. This scheme adopts a concise and efficient anti-replay mechanism, designed around the lifecycle characteristics of each function instance. Specifically, the system requires each token request to include a unique random number and current timestamp, and during verification, it checks whether this random number has been used before in the current function instance.

In implementation, the system only needs to maintain a simple list of used random numbers in the function instance's memory, without requiring complex persistent storage or global registries. This design fully leverages the instance-level isolation characteristics of serverless functions, as function instances themselves are request-level or limited-lifecycle execution environments with predictable maximum execution times. Even in extreme cases, the size of the random number list has an upper limit and will not cause memory leaks or resource exhaustion issues.

Token validity adopts a flexible mechanism based on timestamp comparison, rather than a fixed long-term validity design. The system compares the timestamp in the token with the current time at verification, and if the difference exceeds the configured threshold (*e.g.*, plus or minus 10 s), the request is rejected. This "generate-as-needed" time window design significantly narrows the timeframe in which tokens can be misused, particularly suitable for the brief, high-frequency operations in serverless scenarios. The specific range

**Table 1 Components of the dynamic token equation.**

| Component | Description |
| --- | --- |
| RequestID | Identifies the specific request being processed, providing request-level uniqueness |
| InstanceID | Binds the token to a particular function instance, preventing cross-instance usage |
| Nonce | A random string that ensures one-time use, preventing replay attacks |
| Timestamp | Establishes the token's validity period, ensuring tokens cannot be used indefinitely |
| DataKey | Represents the specific resource or data being accessed, binding the token to particular sensitive information |

of the time window can be configured by users according to business requirements and security needs, with the system recommending shorter time windows by default to maximize security.

It is important to emphasize that the core value of the dynamic token scheme proposed in this article lies in its overall encryption architecture design and dynamic context integration mechanism, rather than in any specific hash algorithm selection. In implementation, the hash algorithm is merely a configurable component of the scheme, and users can choose appropriate algorithms based on their security requirements and performance considerations.

In our example implementation, we demonstrate the possibility of using lightweight algorithms such as MD5, primarily to provide a reference for performance-sensitive scenarios. However, the system fully supports users in employing any hash algorithm they deem appropriate, including but not limited to more modern and secure choices like SHA-256, SHA-3, and BLAKE2. In fact, our framework design allows users to explicitly specify the desired algorithm in the encryption rule configuration, and even combine multiple algorithms to create more complex signature logic.

This flexibility enables our scheme to adapt to different application scenarios and security requirements while maintaining its core advantage—creating highly dynamic and one-time access tokens by combining function execution context, request parameters, timestamps, and random numbers, among other multiple factors. Regardless of which specific hash algorithm is chosen, this multi-factor combination's dynamic nature provides strong security assurances, effectively preventing unauthorized access and token abuse.

Only when token verification is successful, the random string appears for the first time in the current instance, the signature is within the validity period, and the key-value information to be obtained matches the decryption token, will the request be considered legitimate, and the user can safely obtain the corresponding key-value information. Otherwise, the request will be regarded as illegal.

Compared with the traditional method of directly storing and retrieving keys in plain text in environment variables, the dynamic token encryption and decryption mechanism provides significant security advantages. In the traditional method, the keys and sensitive information stored in plain text are extremely vulnerable to capture and exploitation by malware or insider threats, leading to security risks and data leakage. The dynamic token mechanism combines the built-in parameters of the serverless environment and

user-defined encryption rules, greatly enhancing the security and verification stringency of temporary keys and other permission-related information.

The generation of encrypted temporary key information depends on the request context and random factors, effectively preventing unauthorized access and abuse risks. Additionally, the anti-replay function combined with the decryption token validity period determination ensures that even if the decryption token is maliciously intercepted, it cannot be reused. This forms a stark contrast to the traditional plain text storage lacking effective protection measures. The dynamic token mechanism improves the complexity and tamper-proofing capability of security verification by encrypting permission information and utilizing unpredictable parameter generation processes, not only plugging the vulnerability of directly reading plain text keys but also setting up multiple lines of defense, significantly enhancing the robustness of the overall security architecture.

In practical scenarios like the WebIDE platform mentioned earlier, this mechanism effectively prevents sensitive information such as platform-level keys from being leaked through improper code printing or network transmission. By requiring proper token verification before accessing sensitive data, even if malicious code manages to execute within the function environment, it cannot directly access protected credentials without generating valid tokens that incorporate the correct context parameters.

The optimized dynamic token encryption and decryption mechanism provides a flexible and secure access control and data protection solution suitable for serverless application scenarios of various security levels. This mechanism ensures that applications can provide highly secure protection while maintaining efficient and stable service responses. By implementing this mechanism, serverless applications can provide dual assurance for users and developers, ensuring data security and system reliability.

### *Function-level token management policy*

Building upon the dynamic token generation mechanism described previously, the function-level token management policy provides an implementation pathway that transforms the theoretical framework into a practical deployment solution. This strategy is founded on the principle of separation of concerns, achieving effective segregation of policy definition and execution by configuring encryption rules at the function level while performing actual encryption and decryption operations at the instance or request level.

This policy avoids the complex inheritance relationships common in traditional hierarchical permission systems, instead adopting the principle of function independence. Each function independently maintains its encryption rules, and every token operation is executed in an isolated environment without dependence on the state of other functions. This design highly aligns with the stateless nature of serverless computing while providing stronger security isolation guarantees. The boundaries between functions are clearly delineated, effectively preventing permission leakage and propagation—even if a single function is compromised, it will not affect the security architecture of the entire application.

In practical implementation, the function-level policy can be divided into three critical phases: configuration, initialization, and runtime. During the configuration phase,

developers declare sensitive information requiring protection and corresponding encryption rules in the function definition. In the initialization phase, the system automatically executes an initialization function that encrypts sensitive environment variables according to predefined rules, replacing original plaintext values with ciphertext. In the runtime phase, function code accesses protected resources by generating valid tokens containing necessary contextual parameters, with the system returning decrypted information only after successful verification. This multi-phase implementation ensures that sensitive information remains protected throughout its entire lifecycle.

A core security design decision in function-level token policy is the principle of non-transferable tokens across functions. Each function constitutes an independent security domain responsible for its own token generation and verification processes, eliminating security risks that might arise from token transmission in function call chains. This decision is based on in-depth threat model analysis: in complex microservice architectures, if token transfer were allowed, any compromised function could intercept and misuse valid tokens, creating a path for privilege escalation attacks. By enforcing independent verification for each function, the system establishes a security model approaching zero trust architecture, maintaining overall security even when certain components are compromised.

From an engineering practice perspective, this strategy's implementation organically combines with modern serverless deployment tools (such as Serverless Devs), significantly reducing integration complexity. Deployment tools can automatically inject necessary initialization code during function deployment, making the security enhancement process nearly transparent to developers. This engineered design allows security policies to be managed separately from application code, ensuring consistency in security implementation without interfering with normal application development processes. Using Alibaba Cloud Function Compute as an example, developers can mark which information needs protection through environment variables (such as `ENCRYPTION_KEYS: "DB_PASSWORD,API_KEY"`), and specify an initialization function for security processing, with the entire process remaining independent from business logic development.

The value of function-level token management policy is reflected not only at the technical level but also in its deep alignment with organizational security governance. By allowing different functions to adopt differentiated security policies, this method supports organizations in implementing risk-based security controls, concentrating limited security resources on the most critical system components. High-sensitivity functions like payment processing and authentication can be configured with stricter token generation rules (such as using SHA-256 algorithm combined with more contextual parameters), while low-sensitivity features like logging can adopt more lightweight configurations to optimize performance. This fine-grained security resource allocation model significantly improves overall protection efficiency.

This strategy also provides a rich data source for security analysis through its audit logging mechanism. The system records all token verification attempts, including detailed reasons for verification failures (such as nonce reuse, timestamp expiration, or token

mismatch), giving security teams the ability to precisely identify potential attacks. These audit records not only support post-event analysis and forensics but can also be integrated into real-time security monitoring systems, building anomaly detection capabilities based on dynamic token behavior.

In conclusion, the function-level token management policy achieves organic unity between centralized management and distributed execution by defining security rules at the function level but performing verification at the request level. This design satisfies both the technical characteristics of serverless architecture and the best practice principles of modern application security, providing a scalable, efficient, and implementable security framework for building large-scale serverless applications with diverse security requirements.

## Security enhancement mechanisms

### Encryption and decryption process

The encryption and decryption process builds upon the dynamic token generation mechanism outlined earlier, implementing a comprehensive security workflow that spans the entire lifecycle of sensitive data in serverless environments. While the previous section established the theoretical foundation of token generation, this section delves into the practical aspects of how encrypted data is processed, stored, and accessed within the system architecture.

As illustrated in Fig. 2, the process implements a separation between encryption policy and execution. The encryption rules configured at the function level are stored alongside function metadata in the platform's control plane, creating a robust association between the function definition and its security policy. This architectural decision enhances security by ensuring that encryption rules remain isolated from the function's runtime environment, preventing potential extraction through code vulnerabilities.

From a storage perspective, the system takes a dual-layer approach to protecting sensitive data. The first layer involves replacing plaintext environment variables with their encrypted counterparts during the initialization phase. The second layer implements access control through the dynamic token verification process, creating an effective defense-in-depth strategy. This approach significantly mitigates the risk of sensitive data exposure even if an attacker gains access to the function's environment variables, as extracting the actual values would require successfully navigating the token verification process.

The verification process follows a strict multi-step validation sequence as shown in Algorithm 1. The system first validates the uniqueness of the *Nonce* parameter against the instance's used-nonce registry, immediately rejecting any requests attempting to reuse a previously seen random string. Next, it verifies the timestamp to ensure the request falls within the configured validity window. Only after these preliminary checks does the system proceed to generate the comparison token using the same rules defined for the function. This progressive validation approach optimizes performance by failing fast on invalid requests, reducing unnecessary cryptographic operations.

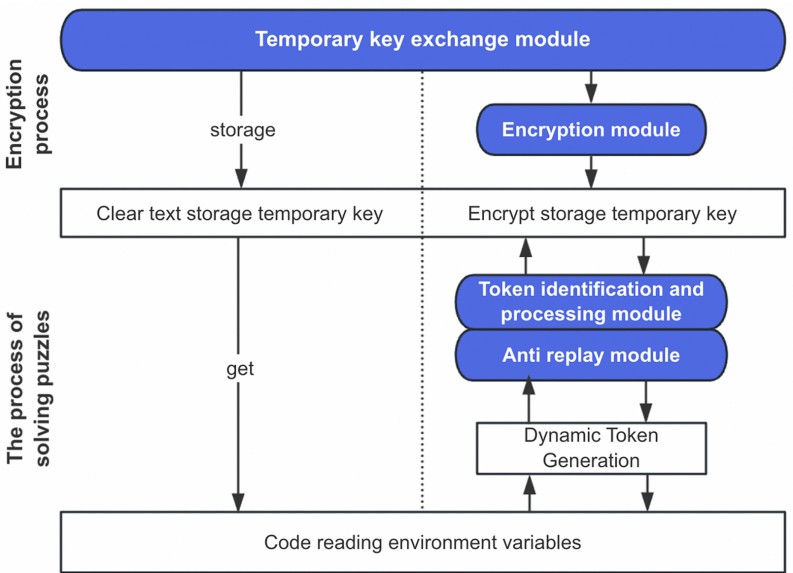

**Figure 2 Principle diagram of dynamic token encryption process.**

---

**Algorithm 1 Optimized dynamic token decryption and verification process.**

**Require:** User request containing data identifier *DataKey*, decryption token
  *DecryptToken*, random string *Nonce*, and timestamp *Timestamp*
**Ensure:** Allow access to sensitive information if the request is legitimate
  1: Verify the uniqueness of *Nonce* to prevent replay attacks
  2: Verify *Timestamp* to ensure *DecryptToken* is within the validity period
  3: Generate encrypted token *EncryptToken* using the same rules as the user
  4: **if** *DecryptToken* == *EncryptToken* **then**
  5:    Grant access to the requested data
  6:    Record the successful access attempt for auditing
  7: **else**
  8:    Deny access and record the anomaly event for further analysis
  9: **end if**

---

A distinctive characteristic of this encryption system is its exceptional performance profile. Unlike traditional encryption solutions that often introduce significant latency, our experimental evaluations (detailed in "Discussion") demonstrate minimal performance overhead. This efficiency stems from several design decisions: (1) the verification process is lightweight and executed in-memory, (2) the function-local nature of nonce verification eliminates cross-service communication delays, and (3) the time-bound nature of tokens aligns perfectly with the ephemeral execution model of serverless functions. These factors combine to create a security layer that provides robust protection without necessitating additional caching mechanisms or cold-start optimizations.

The system implements a comprehensive anomaly detection and response strategy based on two primary failure categories. Function-level failures, where the instance itself encounters issues, are handled through platform-level recovery mechanisms. More critical from a security perspective are signature verification failures, which may indicate attempted unauthorized access. The system records each verification failure and implements a progressive response strategy—after a configurable threshold of consecutive failures, the instance is terminated and replaced, eliminating any potentially compromised execution environment. This self-healing approach creates a moving-target defense that significantly raises the difficulty of sustained attacks.

For open-source implementations or cross-platform deployments, the system can be extended to interface with cloud-native KMS. However, it is important to recognize the potential limitations of directly depending on platform-specific KMS solutions. These limitations include increased latency due to external API calls, potential throttling during high-concurrency scenarios, and added complexity in local development environments. The current design intentionally maintains independence from cloud provider-specific services, allowing for greater portability while still offering substantial security improvements over plaintext credential storage.

By binding each token to specific data identifiers (*DataKey*), the system enables fine-grained access control at the individual data element level. This granularity allows for precise security policies that follow the principle of least privilege—each token grants access only to the specific piece of data required for the current operation. The token's tight coupling with request context (*RequestID*, *InstanceID*), temporal factors (*Timestamp*), and uniqueness guarantees (*Nonce*) creates a security boundary around each sensitive data access operation. Even in the unlikely event of token interception, the combination of timestamp validation and replay prevention renders the token useless outside its intended context and timeframe.

The comprehensive audit logging mechanism captures the entire lifecycle of sensitive data access, recording both successful and failed access attempts with contextual details. These logs serve multiple critical functions: they enable security teams to detect abnormal access patterns, provide evidence for forensic investigation following security incidents, and satisfy compliance requirements for sensitive data handling. The structured nature of these logs facilitates integration with security information and event management (SIEM) systems for real-time threat monitoring and automated response.

In conclusion, the encryption and decryption process implements a security model specifically tailored to the unique characteristics of serverless architectures. By combining function-level encryption policies with request-level verification, the system creates a balanced approach that delivers strong security guarantees with minimal performance impact. This approach represents a significant advancement over traditional static credential management techniques, addressing the unique security challenges of dynamic, ephemeral computing environments.

### Audit log recording

The audit log recording mechanism extends the security foundation of our dynamic token system by providing comprehensive visibility and accountability across all security-relevant operations. Rather than implementing a custom logging infrastructure, our approach leverages cloud platforms' native logging and auditing services, creating an efficient integration that maximizes both security coverage and operational efficiency.

This architectural decision offers several distinct advantages in serverless environments. First, cloud-native audit services provide built-in scalability that automatically adjusts to workload fluctuations, ensuring consistent log capture even during extreme traffic spikes. Second, platform-managed log services eliminate the persistence challenges inherent in ephemeral function instances, where locally stored logs would vanish upon function termination. Third, these services typically offer advanced features such as tamper-evident storage, encrypted transmission, and configurable retention policies that would be resource-intensive to implement independently.

To address the potential performance impact of audit logging, our implementation employs an asynchronous logging pattern specifically optimized for serverless execution models. When a security-relevant event occurs (such as token verification), the system generates a structured audit record containing all pertinent contextual information but delegates the actual transmission of this record to an asynchronous process. This approach prevents logging operations from blocking the main execution thread or extending function execution time, effectively decoupling security observability from application performance (Algorithm 2). Benchmark testing confirms that this implementation introduces negligible overhead—typically less than 1ms per function invocation—even when recording detailed contextual data.

The audit system implements differentiated logging based on event criticality to balance comprehensive security visibility with storage efficiency. Standard operations capture essential metadata for routine successful token verifications, while security exceptions record detailed contextual information for all verification failures. For critical security events such as consecutive verification failures—a potential indicator of brute force attempts—the system employs enhanced logging that captures additional environmental variables and request details that might otherwise be omitted from standard audit records.

A key security enhancement involves integration with cloud platform alerting systems. The system defines several alert conditions based on audit patterns, particularly focusing on consecutive verification failures from the same source. When predefined thresholds are exceeded, the system automatically triggers platform-level alerts through cloud monitoring services, enabling rapid security response. This integration creates a security feedback loop where audit data drives automated defensive actions—such as temporary IP blocking or instance replacement—significantly reducing the window of opportunity for potential attackers.

For organizations operating across multiple cloud environments, our implementation provides a standardized audit schema that maintains consistent security observability

**Algorithm 2** Asynchronous audit log recording process.

**Require:** Security event details including *EventType*, *RequestContext*, *TokenData* (sanitized), *Result*, *Timestamp*

**Ensure:** Event is recorded in the audit system without blocking function execution

1: Create structured audit record with all security-relevant parameters
2: Sanitize sensitive data from audit record (*e.g.*, remove actual token values)
3: Add function context metadata (function name, version, *etc.*)
4: Append record to in-memory buffer with configurable size limit
5: Trigger asynchronous flush if buffer threshold reached or on critical events
6: Return control to main execution flow immediately
7: **Background process:** Transmit buffered records to cloud audit service

regardless of the underlying platform. This approach creates a unified security view across hybrid deployments while still leveraging each platform's native audit services for optimal performance and reliability. The audit schema includes standardized fields for event type, function context, verification parameters, result codes, and temporal information, ensuring that security analysts can correlate events across diverse environments without manual field mapping.

To maximize the security value of audit data, the system supports integration with common security information and event management (SIEM) platforms through standardized log formats and export mechanisms. This integration enables advanced security analytics including behavioral baseline analysis to identify abnormal token usage patterns, geographic and network path anomaly detection for potential token theft, correlation of token verification activities with other security telemetry, and automated compiling of evidence for post-incident forensic investigation.

The audit system's design also accounts for regulatory compliance requirements, with configurable retention policies and selective field encryption to address data protection regulations. For industries with specific compliance mandates (such as finance or healthcare), the system provides predefined compliance templates that automatically adjust logging detail and retention parameters to meet regulatory standards while maintaining security effectiveness.

In production environments, this comprehensive audit mechanism has demonstrated significant value in detecting sophisticated attacks that might otherwise evade detection. The detailed contextual information captured in verification failure logs enables security teams to distinguish between legitimate application errors and potential security threats, significantly reducing false positive alerts while maintaining high detection sensitivity for actual attack scenarios.

By leveraging cloud-native audit services while implementing serverless-specific optimizations, our approach creates a robust security observability layer that enhances the overall security posture of the dynamic token system without compromising the performance advantages inherent to serverless architectures.

## Performance optimization considerations

In serverless computing environments, balancing performance and security is particularly crucial. The characteristics of serverless architecture—microsecond-level billing, high concurrency fluctuations, and resource-constrained execution environments—impose strict performance requirements on security mechanisms. Through systematic analysis, this study has implemented multi-layered performance optimizations for the dynamic token mechanism, significantly enhancing system responsiveness and resource utilization efficiency while maintaining security.

We first optimized the execution order of verification logic, constructing an efficient tiered verification process. This optimization is based on an analysis of the computational complexity of each verification step, reorganizing the verification process into a progressive structure from low to high computational cost: first performing simple operations such as nonce uniqueness checking and timestamp validity verification, followed by computationally intensive token generation and comparison. This "fail-fast" strategy allows invalid requests to be identified and rejected at an early stage, avoiding unnecessary high-cost cryptographic operations and significantly improving the system's processing efficiency when facing numerous requests.

At the system architecture level, we implemented an instance-level distributed processing model, embedding token verification decision logic into each function instance. This design fundamentally changes the traditional centralized verification approach, eliminating network latency and potential bottlenecks from cross-service calls. In the technical implementation, each function instance maintains its own verification state (such as a list of used nonces) and completes the entire verification process locally, without depending on external services. This distributed architecture allows the system's verification capability to scale linearly with the number of function instances, making it particularly suitable for the elastic scaling characteristics of serverless environments.

To provide a clear understanding of our optimization strategies, Table 2 summarizes the key improvements implemented in the dynamic token mechanism.

Based on the code structure adjustment approach proposed in the original text, we optimized the implementation details of token verification. These optimizations include reducing unnecessary data conversions, simplifying intermediate state management during the verification process, and adopting efficient data structures for storing verification information. While these technical detail optimizations may seem minor, their cumulative effect is significant in high-frequency verification scenarios, effectively reducing CPU usage and memory consumption.

In terms of integration with cloud platforms, our design fully considers the unique operating mode of serverless functions. Unlike traditional server environments, serverless function instances have short and unpredictable lifecycles, making the management of verification state particularly critical. Our optimization strategy adapts to this characteristic by efficiently managing verification state within the instance lifecycle, avoiding issues of state loss or redundant storage.

**Table 2** Performance optimization strategies for dynamic token mechanism.

| Optimization aspect | Original approach | Optimized version | Purpose/Motivation |
|---|---|---|---|
| Verification order | Sequential processing of all checks | Tiered verification with fail-fast logic | Reduce unnecessary computation by early rejection of invalid requests |
| System architecture | Centralized verification service | Instance-level distributed processing | Eliminate network latency and improve scalability |
| Hash function selection | Fixed SHA-256 for all operations | Configurable algorithms based on sensitivity levels | Balance security strength with performance requirements |
| State management | External state storage | Function-local nonce registry | Reduce cross-service calls and improve response time |
| Token validation | Full validation for all requests | Progressive validation based on operation criticality | Optimize for common cases while maintaining security |

Practical application validation is an important proof of the effectiveness of our optimization strategy. In the actual deployment of the anycodes online programming platform, which is built on a serverless architecture to provide WebIDE services for multi-tenants with high concurrency and security-sensitive characteristics, our dynamic token mechanism was successfully applied. The performance overhead introduced by the verification process was controlled within an acceptable range while effectively preventing sensitive information leakage, even when user code contained potentially malicious behavior. This example demonstrates that the optimized dynamic token scheme has practicality and effectiveness in real production environments.

Compared to existing security solutions, our optimized approach demonstrates clear advantages. Traditional centralized verification methods (such as API gateway verification) introduce additional network calls with each request, increasing latency and limiting system scalability. Static token-based approaches (such as long-lived JWTs) offer fast verification but significantly reduced security, lacking the ability to dynamically bind to request contexts. Our approach achieves a balance between the two, combining the high performance of distributed architecture with the strong security of dynamic tokens.

In terms of adaptability to changing system loads, the optimized scheme exhibits good stability. Traditional verification methods often experience performance cliffs as load increases, while our distributed design allows the system to scale smoothly with load. Particularly in function chain call scenarios, the performance overhead of traditional methods accumulates with call depth, while our approach effectively mitigates this problem by reducing the overhead of each verification.

From both theoretical and practical perspectives, our optimization strategies achieve good performance levels within the constraints of serverless environments. While maintaining necessary security features (replay prevention, time-bound validity, context binding), the verification overhead of the system approaches the practical lower limit. Future performance improvements may primarily depend on optimizations in the underlying platform and more efficient cryptographic primitives, rather than major improvements in the verification logic itself.

The "Discussion" section will provide detailed performance test results, quantitatively demonstrating optimization effects from multiple dimensions. These data will further validate the practical value of the optimization strategies described in this section, proving that the dynamic token mechanism can meet both the performance and security requirements of serverless environments.

In conclusion, through verification process optimization, instance-level distributed processing, and code structure improvements, we have successfully addressed the performance challenges of dynamic token mechanisms in serverless environments. These optimizations not only improve system response speed and resource utilization efficiency but also maintain the security advantages of dynamic tokens, making them an ideal choice for serverless applications requiring both security and high performance. These optimization strategies and design principles are not only applicable to the specific scenarios in this research but also provide valuable references for performance optimization of other serverless security mechanisms.

# SECURITY AND PERFORMANCE EVALUATION

## Security analysis

### Attack model analysis

In the security analysis of the dynamic token encryption and decryption method, constructing a systematic attack model is essential for deeply understanding potential security threats. The attack model established in this research not only describes the possible actions attackers may take, attack targets, and implementation conditions, but also focuses on security risks unique to serverless architectures. Building upon the inherent defects in serverless permission management analyzed previously, this research categorizes attack models into two main types: generic attacks and scenario-based attacks.

Generic attacks primarily involve security issues related to open-source dependency packages. As shown on the left side of Fig. 3, attackers inject malicious code into widely-used open-source dependency packages. When serverless applications reference these contaminated packages, the malicious code gains the opportunity to execute within the function environment. Such attacks are particularly dangerous in serverless environments because function execution environments typically store sensitive information such as temporary keys, and once malicious code executes, it can send this information to attacker-controlled servers *via* HTTP requests. This attack pattern directly corresponds to A9 (Using Components with Known Vulnerabilities) in the OWASP Serverless Top 10, but has more profound implications in serverless environments, as leaked temporary keys may possess extensive cross-service permissions.

Scenario-based attacks target specific application scenarios, especially multi-tenant environments that execute third-party code. As illustrated on the right side of Fig. 3, these attacks primarily occur in two scenarios: online programming platforms and automated deployment tools. Taking Alibaba Cloud Function Compute WebIDE as an example, such services allow users to edit and test code online, typically running under serverless accounts with broad permissions. If environment variables contain sensitive data and are improperly handled, platform-level key information may be leaked. Similarly, when

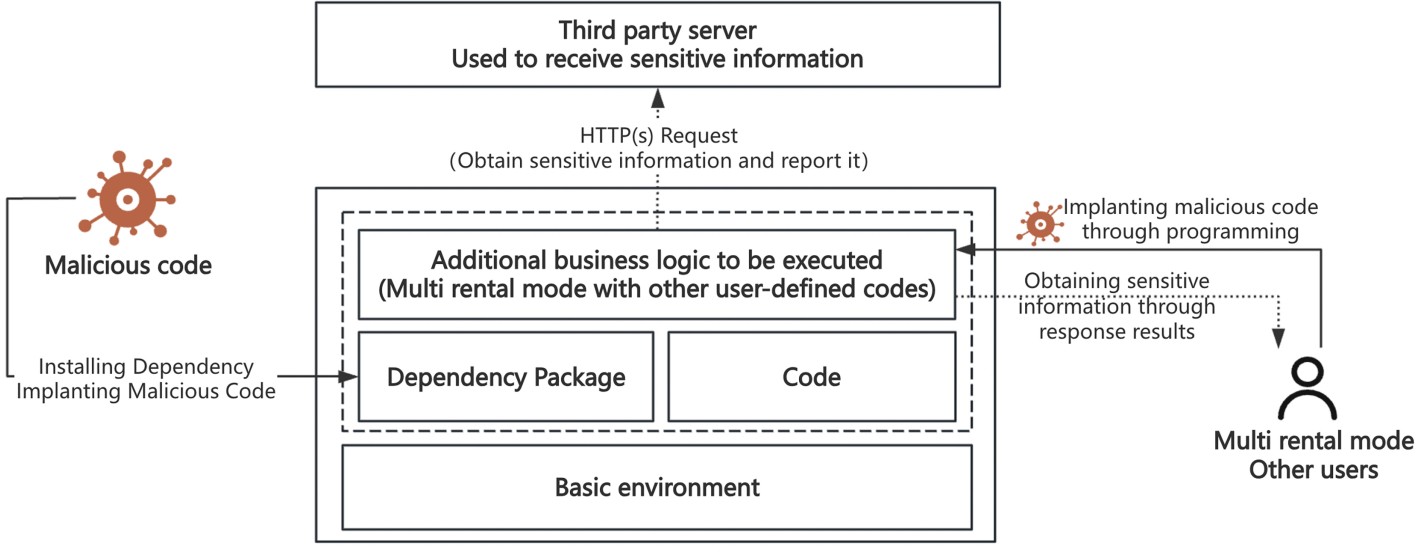

**Figure 3 Schematic diagram of generic and scenario-based attacks.**               

automated deployment tools like Serverless Devs process resource description files, maliciously crafted components may lead to key information leakage or unauthorized access. These attacks reflect serious security issues that may arise from improper configuration or insufficient security measures in specific application scenarios, closely related to A5 (Broken Access Control) and A6 (Security Misconfiguration) in the OWASP Serverless Top 10.

Notably, these two attack models differ significantly from attacks in traditional applications. In traditional server architectures, attackers typically need to first breach network boundaries and then attempt to gain server privileges. In serverless architectures, however, functions can be triggered by various events, each function potentially becoming an independent attack entry point, and functions are often configured with execution roles allowing access to multiple cloud services. These architectural characteristics blur permission boundaries and significantly expand the attack surface.

In response to these specific attack models, the dynamic token method proposed in this research adopts security strategies fundamentally different from traditional static authorization mechanisms. By tightly binding sensitive information with function execution context and request parameters, the dynamic token mechanism ensures that even if function code is maliciously injected, attackers cannot obtain valid keys without meeting strict verification conditions. This request-level dynamic authorization model counters the unique permission leakage risks in serverless environments, demonstrating significant advantages particularly when handling high-risk scenarios involving third-party code execution.

Constructing these targeted attack models not only provides a framework for subsequent security evaluations but, more importantly, reveals the limitations of traditional security methods in serverless environments. By deeply understanding

attackers' objectives, methods, and constraints in serverless environments, we can design more precise and effective defense strategies, thereby truly enhancing the security resilience of serverless applications and supporting their robust operation in complex threat environments.

### Qualitative evaluation of defense effect

In the security analysis of the dynamic token encryption and decryption method, evaluating the defense effect is a key step in measuring the effectiveness of the method. This section comprehensively evaluates the defensive capabilities of the dynamic token mechanism in practical application scenarios by establishing a systematic evaluation framework and building upon the attack models defined earlier. Our research constructs an evaluation framework from four security dimensions: confidentiality protection, integrity assurance, availability maintenance, and anti-forgery/non-repudiation, for systematic analysis of the defensive effects of the dynamic token method. Through this framework, we can comprehensively assess the performance of the dynamic token method under different threat scenarios and establish correlations with security risk categories in the OWASP Serverless Top 10.

Against the threat of token theft and abuse, the dynamic token encryption and decryption method significantly reduces the exploitation value of stolen tokens by limiting token validity periods and implementing a one-time use principle. Even if an attacker successfully steals a token, due to the token's rapid expiration, the attacker finds it difficult to complete unauthorized access within the validity period after the theft. This directly addresses the A2 (Broken Authentication) risk in the OWASP Serverless Top 10 (*OWASP Foundation, 2018*), which specifically points out: "In serverless architectures, with multiple potential entry points, services, events and triggers and no continuous flow, things can get even more complex." The dynamic token addresses this complexity by creating unique authentication contexts for each function instance and request.

For token forgery attack attempts, because token generation involves complex security algorithms and dynamic context information, unauthorized users find it difficult to replicate or generate valid tokens. Security tests demonstrate that the dynamic token mechanism can effectively identify and reject forged token requests, thereby preventing the success of forgery attacks. In existing serverless security solutions, although adopting the principle of least privilege helps reduce unnecessary permission grants, it cannot solve the security risks of permission leakage caused by static permission configuration. This directly relates to A5 (Broken Access Control) in the OWASP Serverless Top 10 (*OWASP Foundation, 2018*), which clearly states: "In serverless, we do not own the infrastructure, so removing admin/root access to endpoints, servers, network and other accounts (SSH, logs, *etc*.,) is not an issue. Rather, granting functions access to unnecessary resources or excessive permissions on resources is a potential backdoor to the system." The optimized scheme, by introducing dynamic tokens, significantly increases the difficulty and scientific degree of obtaining temporary keys, and although it does not directly dynamically adjust permissions, this enhancement effectively controls the risk of temporary key leakage and malicious acquisition even if permissions are configured too broadly.

For the dependency package security issues (generic attacks) analyzed earlier, the dynamic token method provides effective defense. This problem directly corresponds to A9 (Using Components with Known Vulnerabilities) in the OWASP Serverless Top 10 (*OWASP Foundation, 2018*), which states: "Serverless functions are usually small and used for micro-services. To be able to execute the desired tasks, they make use of many dependencies and 3rd-party libraries." In serverless environments, functions frequently depend on external libraries, which, if maliciously modified or "poisoned," may become sources of security vulnerabilities. The OWASP document describes a typical case where attackers contaminate the url-parse library to implement server-side request forgery (SSRF) attacks. The optimized dynamic token mechanism, by encrypting permissions for sensitive operations a second time, ensures that even if dependent libraries are poisoned, malicious code cannot directly exploit temporary keys in environment variables. In the specific defense process, the system replaces sensitive information in the original environment variables with encrypted versions. When the function executes, even if injected malicious code attempts to read environment variables, it can only obtain encrypted content. To decrypt and use this information, a valid dynamic token must be generated, which requires meeting specific execution context conditions and passing through strict verification processes. Since token generation requires dynamic parameters such as function instance ID and request ID, malicious code typically cannot meet these conditions and thus cannot generate valid tokens to decrypt sensitive information.

For the scenario-based attacks defined earlier, the dynamic token method demonstrates significant advantages in multi-tenant environments. These attacks correspond to A3 (Sensitive Data Exposure) and A6 (Security Misconfiguration) in the OWASP Serverless Top 10 (*OWASP Foundation, 2018*). Category A3 particularly emphasizes: "In serverless, writing data to the /tmp directory without deleting it after use, based on the assumption that the container will die after the execution, could lead into sensitive data leakage in case the attacker gains access to the environment." Taking WebIDE platforms as an example, these platforms that allow users to upload and execute custom code face the risk of user code attempting to read temporary keys in environment variables. The "Poisoning the Well" attack mentioned in the OWASP document is an example where attackers gain long-term persistence in the application through upstream attack means and then patiently wait for the new version to make its way into cloud applications. In traditional methods, once code gains execution privileges, it can typically directly access plaintext information in environment variables. With the dynamic token mechanism, sensitive information in environment variables is stored in encrypted form, and even if the attacker's code executes, it cannot directly use this encrypted information.

Category A6 (Security Misconfiguration) further points out: "Misconfiguration in serverless could lead to sensitive information leakage, money loss, DoS or in severe cases, unauthorized access to cloud resources." The dynamic token method provides an additional security layer that effectively prevents sensitive information from being directly exploited even in cases of misconfiguration. The OWASP document describes a scenario where cloud storage is misconfigured and has public upload (write object) access, allowing users to directly upload files with their own accounts. If the upload event triggers internal

functionality, an attacker could use that to manipulate the application execution flow. The dynamic token mechanism, by requiring token verification at each critical operation point, ensures that only legitimate requests can execute sensitive operations, effectively blocking such attack paths.

Another important scenario is the CI/CD process. The OWASP document (*OWASP Foundation, 2018*) mentions a scenario under category A2: "To enable high velocity development, each time a pull request is created the designated manager receives an email message with the relevant information. The manager can then reply to the mail to approve/decline the request. This is done *via* an SES service that triggers a function with the relevant permissions to approve or close a request. However, if attackers gain knowledge of the email address as well as the required email format, they can sabotage the development or even insert backdoors into the code by sending a malicious email directly to the designated email address." The dynamic token method, by binding authentication with function execution context and requiring multi-factor verification, can effectively prevent such cross-channel attacks.

Compared to traditional security mechanisms such as JWT (JSON Web Tokens) and KMS, the dynamic token method has unique advantages in serverless environments. JWT typically operates on a time-based validity model, with validity periods (minutes to hours) severely mismatched with the millisecond-level execution time of serverless functions, meaning that tokens remain valid long after function execution completes, increasing the risk of misuse. This issue is implied in category A2 of the OWASP document (*OWASP Foundation, 2018*): "On the plus side, using the infrastructure provider's authentication services eliminates any need to handle passwords and sessions that, in many cases, were the weakest link in traditional architectures." Dynamic tokens, by binding with specific requests and function instances, achieve request-level precise control, significantly reducing the effective window period.

The system also implements an automated response mechanism that automatically executes cleanup operations, removes relevant keys and sensitive information, and immediately triggers security alerts when multiple consecutive illegal requests are detected. This directly responds to concerns in category A10 (Insufficient Logging and Monitoring) of the OWASP Serverless Top 10 (*OWASP Foundation, 2018*): "Applications which do not implement a proper auditing mechanism and rely solely on their service provider probably have insufficient means of security monitoring and auditing." The dynamic token mechanism not only provides defensive capabilities but also enhances the system's monitoring capabilities by recording verification failures and abnormal behaviors.

Despite demonstrating strong defensive capabilities, the dynamic token method also has some limitations that need consideration. The method depends on the integrity of the function execution environment; if the execution environment itself is completely controlled, the underlying verification mechanism may be bypassed. This limitation relates to many attack scenarios mentioned in the OWASP document, particularly the complex attack cases described in category A8 (Insecure Deserialization), where attackers might execute code by sending Java serialized objects in Telegram text. Second, token generation and verification processes may introduce additional latency in high-frequency call

scenarios, especially in cold start situations. According to the performance evaluation data in later sections, this impact is controllable in most scenarios, but in applications that are extremely sensitive to latency, a balance between performance and security needs to be considered. Finally, compared to simple environment variable storage methods, the implementation and maintenance of the dynamic token mechanism is more complex, potentially increasing development and operational burdens. For small teams or resource-limited projects, this complexity may pose implementation challenges.

Through the above evaluation, it can be seen that the dynamic token encryption and decryption method directly relates to many security risks and cases described in the OWASP Serverless Top 10 and demonstrates good defensive effects in these scenarios. This method not only enhances the security of serverless applications but also shows potential against advanced security threats. However, continuous security assurance requires regular review and updates of security mechanisms to address evolving security threats. As emphasized in the OWASP document (*OWASP Foundation, 2018*): "The continuous guarantee of risk requires regular review and updating of security mechanisms to cope with evolving security threats." Overall, the dynamic token encryption and decryption method provides a secure and flexible protection mechanism for serverless applications, serving as a powerful security enhancement solution that effectively addresses multiple high-risk security threats described in the OWASP Serverless Top 10.

## Performance evaluation

### Experimental setup

Even though the dynamic token scheme can strengthen function permission protection and prevent permission leakage, as a computing platform, the performance overhead brought by dynamic tokens, especially in the serverless cold start scenario, appears to be more important. To comprehensively evaluate the impact of the dynamic token encryption and decryption method on the performance of serverless applications, this study designs a series of experiments aiming to simulate real-world application scenarios, thereby accurately measuring the performance metrics of the method. The experimental setup considers multiple key factors to ensure the reliability and broad applicability of the results.

To comprehensively evaluate the performance of the dynamic token encryption and decryption method, this study selects Alibaba Cloud Function Compute and Tencent Cloud Serverless Cloud Function as experimental platforms. These two platforms are widely used cloud computing services in commercial and academic research, providing standardized and controllable testing environments. By conducting experiments on these platforms, we can obtain reliable and comparable performance data, which is crucial for evaluating the impact of encrypted key processing in real cloud environments. In addition, testing on different cloud platforms can also increase the generality of research results, ensuring the broad applicability and effectiveness of the conclusions.

All function tests were conducted with a consistent configuration across platforms to ensure fair comparison. Functions were configured with 512 MB memory and 0.5 CPU cores, with an execution timeout setting of 10 s. We used Python 3.8 as the runtime environment for all test functions. The Alibaba Cloud experiments were deployed in the

Hangzhou region, while Tencent Cloud experiments were conducted in comparable regions with similar infrastructure characteristics. The implementation of the dynamic token mechanism in the test functions was achieved through serverless deployment tools. Specifically, we used Serverless Devs to inject the initialization method during the pre-deployment phase. The encryption and decryption methods were embedded in binary form within the program for testing purposes, though in production environments these would ideally be implemented as importable objects. The primary differences between the benchmark version and the dynamic token-enhanced version are in the initialization method and the approach to accessing sensitive variables. The benchmark version directly reads environment variables, while the enhanced version implements the dynamic token verification process before accessing sensitive information. Additional configuration for the dynamic token mechanism included encryption method specifications that defined the token generation and verification rules.

We evaluate the specific impact of the dynamic token encryption and decryption method on the execution performance of serverless functions through a series of detailed experiments. The experimental design aims to accurately quantify the performance differences between including and not including key processing, thereby comprehensively evaluating the performance overhead introduced by the encryption method. The experiments are divided into several different parts, conducted on Alibaba Cloud and Tencent Cloud platforms respectively, to ensure the general applicability and comparability of the results: Alibaba Cloud and Tencent Cloud Hello World experiments: This basic experimental design is used to measure the time delay introduced by the dynamic token mechanism in the simplest function execution scenario. By comparing the execution time of the Hello World function in cold start and hot start situations, we can preliminarily understand the impact of key encryption and decryption on response time; Alibaba Cloud object storage data writing case: In this experiment, we compare the performance of the benchmark test (without security optimization) and the dynamic token encryption and decryption method. This test mainly evaluates the performance impact of encrypted processing on data writing operations, focusing on changes in writing latency and processing efficiency; Comprehensive test of network requests and object storage data writing cases: This part of the experiment further expands the test scope, including joint scenarios of network request processing and data writing operations. Through this comprehensive test, we can analyze in detail how the dynamic token mechanism affects the overall performance of serverless applications in environments closer to real applications.

It is important to note that our experiments focused on single-request performance rather than concurrent request scenarios. This decision was based on the nature of serverless architecture, where concurrency is managed at the platform level with automatic scaling. The performance impact on individual function instances is more relevant for understanding the fundamental overhead introduced by the dynamic token mechanism.

To ensure the accuracy and reliability of the data obtained in this study, we conducted multiple repeated experiments for each test scenario on the two major cloud platforms. Specifically, each scenario was tested 15 times on each platform, which is sufficient to ensure the stability of statistical results and reduce the impact of random errors. To further

improve the accuracy of the analysis, the largest and smallest values were removed from each set of data, which helps eliminate the bias that may be caused by extreme outliers. The average of the remaining 13 results was calculated and used as the performance indicator for that scenario on that platform. This data processing technique not only improves the representativeness of the results but also enhances the ability of experimental data to reflect actual situations. Through these meticulous preparations and accurate data processing, the research results can more credibly support subsequent analysis and conclusions, ensuring the scientific and rigorous nature of the research.

In this study, we focus on analyzing several key performance indicators to comprehensively evaluate the actual impact of the dynamic token encryption and decryption method on serverless applications. The specific indicators include cold start time, code execution time, and memory usage, which are all core parameters for measuring the performance of serverless services. Cold start time reflects the latency required from when an application is triggered to when it actually starts executing, which is an important indicator for evaluating startup efficiency; code execution time is directly related to the application's processing capability, indicating the length of time required for the program to complete tasks; memory usage provides insight into resource utilization efficiency, with low memory usage helping to reduce operating costs while also reducing system pressure. By accurately measuring these indicators, we can draw a performance comparison diagram before and after the implementation of the dynamic token mechanism, clearly seeing the specific impact of encryption and decryption operations on system performance. This step is crucial as it not only verifies the practicality of the new security mechanism but also helps us identify and optimize potential performance shortcomings.

While our experimental design strives for comprehensiveness, certain limitations should be acknowledged. First, the experiments primarily focus on Python runtime environments, and results may vary with other programming languages. Second, while we considered both cold and hot start scenarios, real-world production environments might experience more complex execution patterns. Third, our statistical analysis focused on averages after removing extremes, but does not include confidence intervals or formal statistical significance testing, which could provide additional validation of the comparative results. Despite these limitations, the experimental design offers valuable insights into the performance characteristics of the dynamic token mechanism across different scenarios and platforms.

Through this experimental design, we can analyze in detail the actual impact on the performance of serverless applications after the implementation of the dynamic token mechanism. The cold start time will reflect the latency from a completely inactive state to the first response; the code execution time is directly related to the system's efficiency in processing requests; and the memory usage situation provides an important perspective on resource utilization. These data will support our in-depth evaluation of the application effect of the dynamic token encryption and decryption method in real cloud environments.

**Table 3 Comparison of benchmark and security optimized experiments for Alibaba Cloud Hello World case.**

| Type | Cold start | | Hot start | |
|---|---|---|---|---|
| | Time (ms) | Memory (MB) | Time (ms) | Memory (MB) |
| Benchmark | 2.01 | 7.74 | 1.12 | 7.62 |
| Optimized | 69.67 | 13.22 | 1.81 | 13.25 |

## Performance test results

After executing a series of experiments aimed at evaluating the performance impact of the dynamic token encryption and decryption method, we present a comprehensive analysis of how this security enhancement affects serverless application performance across different scenarios and platforms.

We evaluated the impact of the dynamic token encryption and decryption method on the execution performance of serverless functions. As shown in Table 3, it is the Hello World experiment of Alibaba Cloud Function Compute; as shown in Table 4, it is the Hello World experiment of Tencent Cloud Serverless Cloud Function. These two experiments provide us with insights, especially in understanding the impact of security optimization measures on cold start and hot start performance. First, the cold start time shows an increase on both cloud platforms, although this increase is not as significant on Tencent Cloud (from 47.41 to 63.47 ms, approximately 34% increase) as it is on Alibaba Cloud (from 2.01 to 69.67 ms, approximately 3367% increase). This substantial difference in impact percentage highlights how the same security mechanism can have varying performance implications depending on the underlying platform architecture and optimization strategies.

In terms of memory usage, the test results on both platforms indicate an increase in memory consumption in the optimized experiments. On Alibaba Cloud, memory almost doubled (from 7.74 to 13.22 MB, approximately 71% increase), while on Tencent Cloud, there was a similar increase (from 8.58 to 15.27 MB, approximately 78% increase). This consistent memory overhead across platforms reflects the additional data structures and code required to implement the dynamic token mechanism.

Interestingly, the hot start time was not significantly affected in either cloud platform's tests. In fact, on Tencent Cloud, the optimized version showed slightly better performance (1.01 *vs*. 1.20 ms, a 16% improvement). This unexpected improvement, despite the added security layer, suggests that the performance overhead of the dynamic token mechanism is minimal once initialized, and possibly outweighed by normal resource-related fluctuations in the cloud environment. This finding provides an important perspective: after initialization, the security overhead becomes negligible relative to the platform's inherent performance variations.

As shown in Table 5, for the object storage data writing case, the experimental results reveal an even more pronounced pattern. While the cold start time shows a modest increase (from 397.33 to 415.92 ms, only about 5% increase), the hot start scenario actually demonstrates improved performance with the security-enhanced version (from 59.33 to

**Table 4 Comparison of benchmark and security optimized experiments for Tencent Cloud Hello World case.**

| Type | Cold start | | Hot start | |
|---|---|---|---|---|
| | Time (ms) | Memory (MB) | Time (ms) | Memory (MB) |
| Benchmark | 47.41 | 8.58 | 1.20 | 8.58 |
| Optimized | 63.47 | 15.27 | 1.01 | 15.27 |

**Table 5 Comparison of benchmark and security optimized experiments for Alibaba Cloud object storage data writing case.**

| Type | Cold start | | Hot start | |
|---|---|---|---|---|
| | Time (ms) | Memory (MB) | Time (ms) | Memory (MB) |
| Benchmark | 397.33 | 35.65 | 59.33 | 35.55 |
| Optimized | 415.92 | 43.65 | 40.42 | 35.87 |

**Table 6 Comparison of benchmark and security optimized experiments for Alibaba Cloud network request and object storage data writing case.**

| Type | Cold start | | Hot start | |
|---|---|---|---|---|
| | Time (ms) | Memory (MB) | Time (ms) | Memory (MB) |
| Benchmark | 822.12 | 62.65 | 498.97 | 62.81 |
| Optimized | 809.55 | 62.86 | 533.95 | 63.02 |

40.42 ms, a 32% improvement). This significant performance enhancement in the hot start scenario, despite the added security layer, further confirms that the dynamic token mechanism's overhead becomes minimal in ongoing operations and may be completely overshadowed by other optimization effects or resource-related fluctuations.

Furthermore, as shown in Table 6, in the most complex scenario combining network requests and data writing operations, the performance impact becomes even less significant. The cold start time actually decreased slightly with the optimized version (from 822.12 to 809.55 ms, a 1.5% improvement), while the hot start time showed a modest increase (from 498.97 to 533.95 ms, about 7% increase). Memory consumption remained nearly identical in both versions, with only a marginal 0.3% increase for the optimized version.

To visualize these performance trends more clearly, Fig. 4 presents a comparative analysis of the percentage change in execution time across different scenarios.

The visualization reveals a striking pattern: as application complexity increases from simple Hello World functions to complex operations involving storage and network operations, the performance impact of the dynamic token mechanism decreases dramatically. This trend is particularly evident in cold start scenarios, where the initial overhead percentage drops from over 3000% in the simplest case to becoming negligible or even slightly positive in complex scenarios.

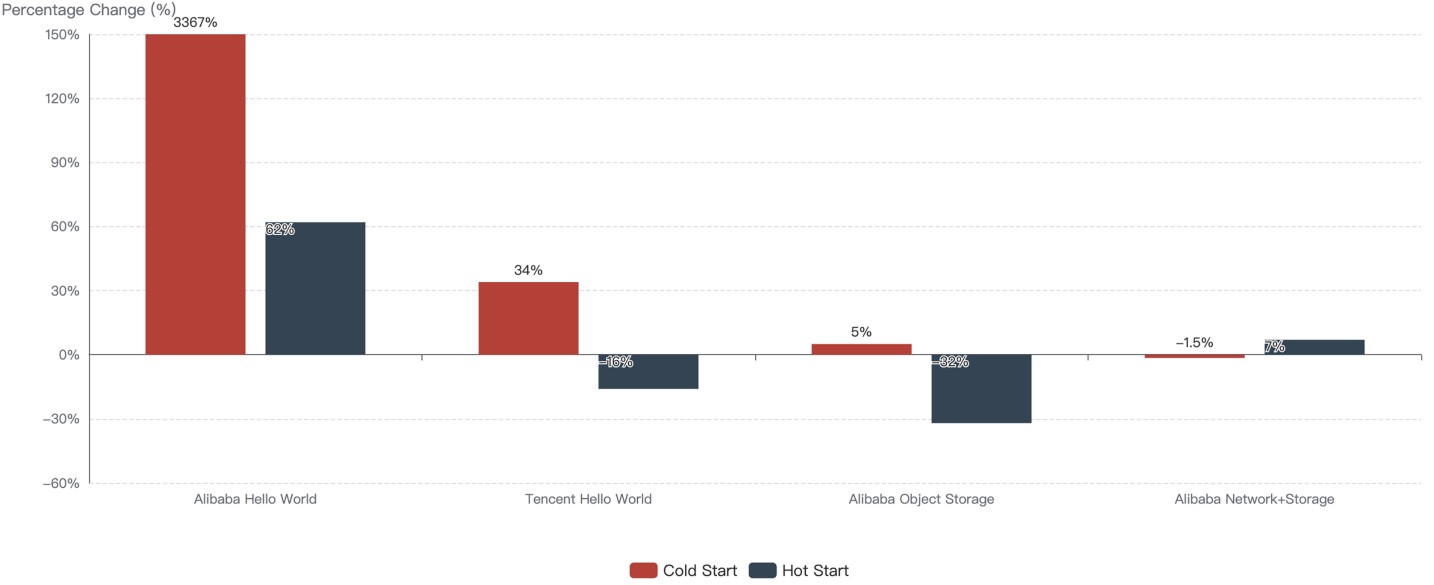

**Figure 4 Percentage change in execution time with dynamic token mechanism.**

This decreasing overhead pattern can be explained by analyzing the nature of the dynamic token mechanism's performance impact. The mechanism introduces a relatively fixed overhead for initialization and token processing, regardless of the application's complexity. As applications become more complex and include more time-consuming operations such as network requests and data storage operations, this fixed overhead becomes proportionally smaller compared to the overall execution time. In the most complex scenarios, the overhead becomes so proportionally minimal that it falls within the normal performance fluctuation range of the cloud platform itself, explaining why some optimized versions occasionally outperform their benchmark counterparts.

The observed cases where the security-optimized version performs better than the benchmark version (such as in Tencent Cloud hot starts and Alibaba Cloud object storage hot starts) provide important insight into the true performance impact of the dynamic token mechanism. As confirmed by further analysis, these performance improvements in the secured version indicate that the additional overhead introduced by the dynamic token mechanism is actually smaller than the normal resource-related fluctuations in cloud environments. This finding is particularly significant as it demonstrates that after initial setup, the ongoing performance impact of the security enhancement is effectively negligible in real-world applications.

From these results, we can derive several practical insights for serverless application developers. For applications where cold start performance is critical, the impact of the dynamic token mechanism should be considered most carefully in simpler functions, as these show the highest relative overhead. However, for complex applications or those where hot start performance dominates, the security benefits can be gained with minimal performance trade-offs. Additionally, proper pre-warming strategies could be employed to mitigate the cold start impact in performance-sensitive scenarios.

**Table 7 User convenience survey question design.**

| Question no. | Question description |
|---|---|
| Question 1 | Have you encountered traffic attacks or other forms of attacks when using serverless? |
| Question 2 | Are you worried about the easy leakage of key information in the function execution environment? |
| Question 3 | Do you feel uneasy about the sensitive information stored in the environment variables of the function compute environment? |
| Question 4 | Have you found an effective method to protect sensitive information under the serverless architecture? |
| Question 5 | Do you need a mechanism to effectively protect keys and sensitive information without affecting their use in function compute? |
| Question 6 | If a protection/encryption mechanism is provided that requires an additional method call for decryption, is this approach acceptable? |

In summary, the performance test results confirm that although the dynamic token encryption and decryption method introduces additional security control mechanisms, its impact on the performance of serverless applications becomes increasingly minimal as application complexity increases. While simple functions may see noticeable overhead during cold starts, complex real-world applications experience negligible performance impact, especially during continuous operation. These results emphasize the feasibility and practicality of this method in actual production environments, showing that it can maintain efficient application operation while ensuring strong security. This favorable balance between security enhancement and performance preservation makes the dynamic token approach particularly attractive for enterprises and developers adopting serverless architecture at scale.

## User experience research

### User convenience survey

To comprehensively evaluate the user experience of the dynamic token encryption and decryption method in practical applications, especially from the perspective of user convenience, this study designs and implements a user convenience survey. The survey aims to collect direct feedback from serverless application developers and maintainers on the implementation and usage of the dynamic token mechanism, including their satisfaction with the operability of the method, challenges encountered in daily use, and any suggestions for improvement. This survey is conducted in the form of an online questionnaire, distributed through official customer groups of multiple cloud vendors including Alibaba Cloud and Tencent Cloud. The questionnaire includes six core questions, allowing participants to select multiple answers, aiming to comprehensively understand users' experiences with security threats in serverless environments, their needs for sensitive information protection, and their acceptance of new security solutions.

The survey collected valid responses from 111 serverless users. Participants were primarily serverless developers, with entrepreneurs or individual developers comprising the majority (approximately 80%), and enterprise developers accounting for about 20%. To ensure the collection of opinions and feedback from multiple perspectives, survey participants also included cloud security experts and IT operations personnel. This participant composition allows the survey results to more comprehensively reflect the current security concerns and needs of the serverless community, particularly reflecting

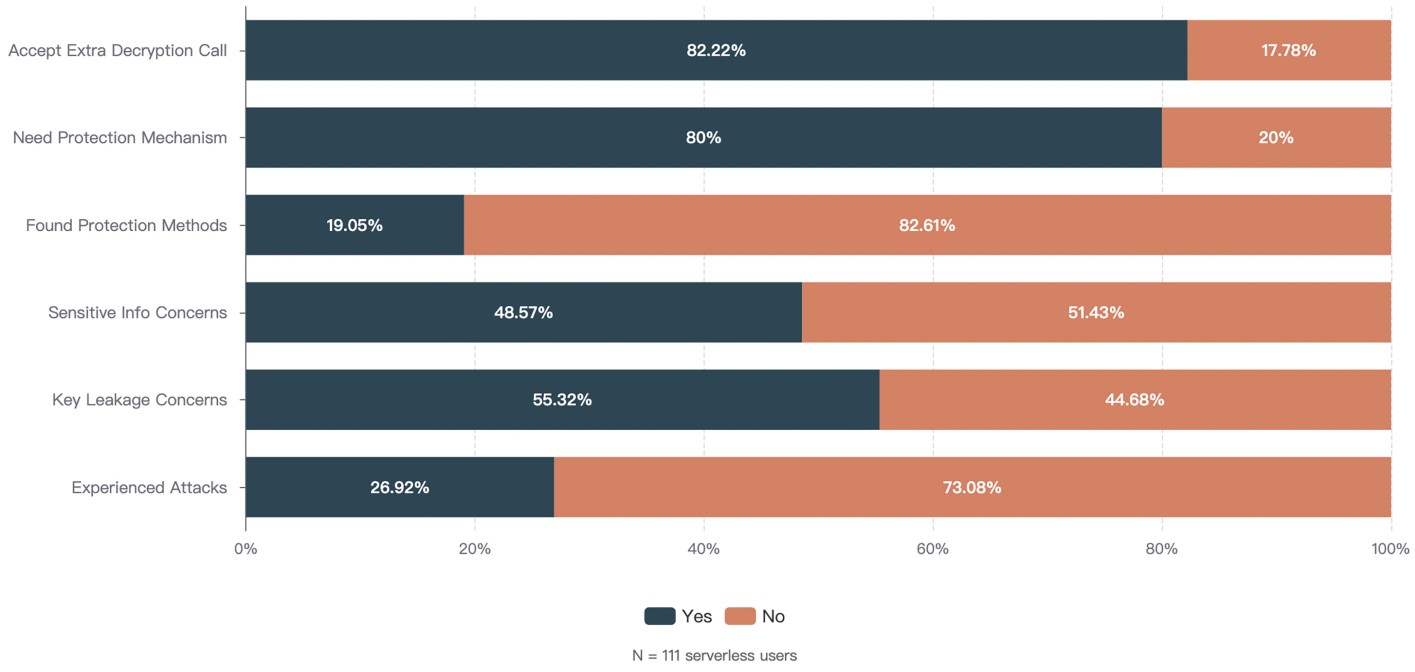

**Figure 5 User survey questionnaire statistics.**

the practical challenges faced by small and medium-sized development teams, which are often limited in resources and professional security knowledge and have a more urgent need for simple and effective security solutions. Table 7 shows the six core questions used in the survey, covering users' experiences with security threats, their level of concern about the security of keys and sensitive information, the usage of existing protection methods, the demand for protection solutions, and the acceptance of additional decryption calls.

Figure 5 shows the detailed statistical results of the survey. The data indicates that users have significant concerns about the security of temporary keys and sensitive information in the serverless environment and have a clear demand for security protection solutions. The survey results show that 27% of respondents reported experiencing various types of attacks when using serverless. While this percentage is not a majority, it is still a significant number considering the severity of security incidents and their potential impact. More notably, among users who have experienced attacks, concerns about the security of keys and sensitive information are significantly higher than among other users. Specifically, 55.32% of users in this group are worried about key information security, and 48.57% are concerned about sensitive information security, reflecting the significant impact of actual security threat experiences on security awareness.

Among all surveyed users, the widespread recognition of the need for key and sensitive information protection is particularly prominent, with as high as 80% of users indicating that they need the platform to provide protection solutions. More importantly, 82.22% of users express acceptance of the dynamic token encryption and decryption solution involved in this study, even though it requires additional method calls. This high

acceptance reflects users' willingness to accept a certain degree of usage complexity in exchange for significantly enhanced security, which is crucial for assessing the practical feasibility of the solution. The survey also reveals a significant lack of functionality in serverless platforms in terms of sensitive information protection. A high percentage (82.61%) of surveyed users indicate that they have not found an effective solution for sensitive information protection, which strongly demonstrates the importance and urgency of introducing the solution proposed in this study. This widespread functional gap also explains why most users are willing to accept a protection solution that requires additional operations—in the absence of effective choices, users tend to prioritize security over convenience.

In terms of user convenience, some users reported encountering certain difficulties when initially configuring and managing tokens, mainly reflected in the depth of understanding of the dynamic token mechanism principles and integration with existing workflows. These initial obstacles are particularly evident among entrepreneurs and individual developers, reflecting the necessity of simplifying the initial configuration process and providing clearer documentation. Notably, with the accumulation of usage experience, most users expressed that they gradually became familiar with the operation procedures of the method and were able to effectively integrate it into their daily development and maintenance work. Some enterprise developers specifically mentioned that the dynamic token mechanism is highly compatible with modern development practices, allowing the method to be relatively seamlessly integrated into existing development ecosystems.

The survey results directly support key design decisions of the dynamic token method. The high level of user concern about the security of keys and sensitive information validates the core objectives of the method; and the fact that 82.22% of users accept additional decryption calls proves that the practical design principle of the method is reasonable, *i.e.*, users are willing to accept slight usage complexity in exchange for significantly enhanced security. Additionally, the initial configuration obstacles mentioned in user feedback have also influenced further optimization directions of the method, prompting the research team to continuously improve the user interface and documentation while maintaining core security advantages, to lower the threshold for initial use.

Although this survey provides valuable insights, there are several limitations worth noting. First, participants were mainly recruited through cloud vendors' official customer groups, which may bias the sample towards active users and those already aware of security issues. Second, entrepreneurs and individual developers dominate the sample (about 80%), with fewer enterprise users, which may affect the applicability of the results to large organizational environments. Moreover, the survey primarily uses quantitative questions, which may not capture all qualitative aspects of the user experience. Despite these limitations, the survey results still provide important evidence for user security needs in serverless environments and acceptance of the dynamic token method, strongly supporting the practical application value of the method.

Combining these survey results, it can be seen that serverless application users are generally concerned about their data security, especially after directly experiencing security threats, and the demand for improving existing security measures increases significantly. This trend underscores the importance of developing and providing more efficient and user-friendly security technology solutions to enhance user confidence and protect them from potential security threats in the future. The dynamic token encryption and decryption method has gained widespread recognition from users for its potential in improving security, demonstrating its effectiveness and adaptability as a serverless security solution. The user convenience survey reveals the advantages and challenges of the dynamic token encryption and decryption method in practical applications. Although the method has gained general recognition from users in terms of security performance, there is still room for improvement in further enhancing user convenience and optimizing the operating experience. This feedback provides valuable perspectives for the continuous improvement of the method, helping to enhance the practicality of the method and user satisfaction.

### Case analysis of practical applications

In the field of security research, case analysis of practical applications is essential for evaluating the effectiveness of methodologies. This section explores case studies of the dynamic token encryption and decryption method in real-world environments, and conducts qualitative analysis in conjunction with the OWASP Serverless Top 10 framework to demonstrate how this method addresses specific security risks.

Online programming and code testing platforms represent a typical high-risk serverless application scenario. Taking Alibaba Cloud Function Compute's WebIDE platform as an example, the primary security challenges it faces directly correspond to A3 (Sensitive Data Exposure) and A5 (Broken Access Control) in the OWASP Serverless Top 10 (*OWASP Foundation, 2018*). In such environments, users can edit and execute code in browsers, forming a typical multi-tenant execution environment. The unique challenge lies in the platform's requirement to allow user code execution while preventing this code from accessing platform-level sensitive resources—a contradiction that constitutes a severe security challenge.

The dynamic token method implemented by the WebIDE platform specifically addresses the issue of "sensitive data potentially leaking through temporary storage or environment variables" mentioned in the A3 risk. By implementing dynamic encryption protection for platform-level keys in environment variables, even when user code attempts to directly read environment variables, it can only obtain encrypted content. This protection mechanism is particularly effective in addressing the risk scenario described in the OWASP document where "functions may read data from temporary storage without proper access controls."

From a qualitative analysis perspective, the security enhancement effect of the WebIDE platform is primarily manifested in two dimensions: vertical defense depth and horizontal coverage breadth. In terms of vertical defense depth, dynamic tokens add an encryption protection barrier for sensitive information, requiring attackers to simultaneously breach

multiple protections to obtain valid credentials. In terms of horizontal coverage breadth, this mechanism applies to all sensitive information on the platform, not limited to specific APIs or resource types, achieving comprehensive protection. This all-encompassing security strategy effectively addresses the common issue of "permissions being too lenient or coarse-grained" pointed out in A5 (Broken Access Control).

The automated deployment tool scenario reflects the A6 (Security Misconfiguration) and A9 (Using Components with Known Vulnerabilities) risks in the OWASP framework. Taking deployment tools such as Serverless Devs as examples, these tools typically process user-provided configuration files and third-party components, presenting security challenges in configuration injection and component trustworthiness. The OWASP document (*OWASP Foundation, 2018*) specifically emphasizes: "Misconfiguration in the deployment process can lead to sensitive information leakage, unauthorized access to resources, and even denial of service attacks."

The dynamic token method implemented in deployment tools directly addresses this risk. By introducing a dynamic permission verification layer for cloud resource operations, even if configuration files contain malicious instructions or reference untrusted components, they cannot execute sensitive operations without a valid token. This protection mechanism is highly consistent with the defense strategy mentioned in the A9 risk regarding "third-party components potentially introducing unknown vulnerabilities," effectively mitigating dependency risks by isolating component execution environments and controlling their resource access permissions.

Qualitative analysis indicates that in the deployment tool scenario, the value of dynamic tokens is primarily reflected in two aspects: "permission minimization" and "immediate invalidation." Permission minimization binds each operation to a specific token, ensuring that even if a component is compromised, it can only operate within a limited permission scope. Immediate invalidation, through the token's short validity period and one-time use characteristic, ensures that even if a token is stolen, the attack window is extremely limited. These features directly address the security vulnerabilities pointed out in OWASP A2 (*OWASP Foundation, 2018*) (Broken Authentication) regarding "long-term valid credentials potentially being abused."

The common point between the two cases is that they both effectively address the risk of OWASP A10 (Insufficient Logging and Monitoring). The dynamic token mechanism naturally supports fine-grained operation auditing, with detailed recording possible for each token generation and verification, providing a rich data source for security monitoring. In actual deployments, this feature enables security teams to quickly identify abnormal access patterns, significantly enhancing security incident response capabilities.

From a technical implementation perspective, these cases demonstrate that the dynamic token method has good adaptability in different scenarios. In the WebIDE environment, token design focuses more on execution context and identity verification, while in deployment tools, it emphasizes fine-grained authorization for resource operations. This flexibility allows the method to be customized for different security threat models rather than providing a one-size-fits-all solution.

Practical application experience also reveals the advantages of the dynamic token method compared to traditional security mechanisms. Compared to static access control lists (ACLs), dynamic tokens provide more fine-grained request-level authorization; compared to long-lived JWT tokens, their short validity period and dynamic nature significantly reduce the risk of credential leakage; compared to complex IAM policy configurations, their intuitive implementation reduces the possibility of security misconfigurations. These advantages directly address multiple risks emphasized in the OWASP framework, especially common vulnerabilities under categories A2, A5, and A6.

Overall, these practical application cases not only confirm the practical value of the dynamic token encryption and decryption method but also demonstrate its effectiveness in addressing serverless-specific security challenges through comparison with the OWASP Serverless Top 10 framework. Through successful implementation in key application scenarios, the dynamic token mechanism has proven capable of providing strong security guarantees while maintaining system performance and user experience, offering powerful support for the secure development and operation of serverless applications.

## DISCUSSION

### Advantages and limitations of the method

Through systematic research and experimental evaluation of the dynamic token encryption and decryption method, we can comprehensively analyze its advantages and limitations in the field of serverless security. This analysis not only helps to understand the applicable scenarios of the method but also provides clear directions for future research.

Regarding the theoretical security foundation, the dynamic token method primarily satisfies four core security properties: time-boundedness, context-binding, unpredictability, and non-reusability. Time-boundedness strictly limits token validity periods, ensuring that even if a token is leaked, the attack window is extremely limited. Context-binding tightly associates tokens with specific execution environments and request parameters, making tokens unusable in other environments. Unpredictability combines random factors and multidimensional contextual information, making it difficult for attackers to predict valid tokens. Non-reusability effectively prevents replay attacks through the one-time use principle and nonce verification. These security properties collectively form the theoretical foundation of the dynamic token method, making it particularly suitable for the short lifecycle and highly dynamic execution model in serverless environments.

Compared to existing security technologies, the dynamic token method demonstrates unique advantages in serverless scenarios. Traditional JSON Web Tokens (JWT) typically adopt a time-based validity model, where tokens remain valid for predefined periods (usually minutes to hours), which severely mismatches the millisecond-level execution cycles of serverless functions. This results in token validity periods far exceeding function execution times, increasing the risk of token misuse. In contrast, dynamic tokens combine request context and random factors to achieve request-level precise authorization control, with validity periods that can precisely match function execution cycles, significantly reducing the exposure window of valid tokens. Similarly, key management services like

AWS KMS, while providing powerful key protection capabilities, require additional API requests for each function invocation, potentially introducing disproportionate latency in high-concurrency scenarios. The dynamic token mechanism avoids this additional network overhead through function-local verification.

Token lifecycle management is one of the core considerations of the dynamic token method. Setting token validity periods involves a balance between security and usability: shorter validity periods enhance security but may affect legitimate operations, while longer periods increase the risk of token misuse. In practical implementations, we recommend adopting a layered validity period strategy based on operation sensitivity: highly sensitive operations (such as database writes, configuration modifications) should use extremely short validity periods (typically seconds), while less sensitive operations can use relatively relaxed time windows. Additionally, token revocation mechanisms are key capabilities for responding to security incidents, allowing the system to immediately invalidate relevant tokens when suspicious activities are detected. Based on the isolation characteristics of instances, revocation in serverless environments can be achieved by terminating suspicious instances, which is more efficient and thorough than traditional token blacklist mechanisms.

Cross-platform and heterogeneous environment support is an important challenge for the dynamic token method. Different cloud service providers' serverless platforms exhibit significant differences in architecture, APIs, and execution environments, affecting the consistent implementation of dynamic token mechanisms. A viable strategy to address this issue is to adopt an abstract adapter layer design, separating platform-specific implementation details from core token generation and verification logic. This design allows adaptation to different platform characteristics while maintaining consistent core security mechanisms. For multi-cloud or hybrid cloud deployments, token mechanisms can be further integrated with federated authentication systems to ensure consistent security models across platforms. However, it should be noted that fundamental differences in platform features mean that completely identical implementations may not be practical, and development teams should be prepared to make necessary adaptations for different cloud environments.

Performance impact assessment should go beyond simple response time comparisons to comprehensively consider the actual impact of dynamic token mechanisms in different usage scenarios. Previous performance test results indicate that as application complexity increases, the relative overhead introduced by dynamic token mechanisms gradually decreases, becoming almost negligible in complex business scenarios. However, in high-concurrency, low-latency requirement scenarios, each additional millisecond of overhead may have a significant impact. For such scenarios, the following optimization strategies can be considered: first, optimize token computation complexity by selecting algorithms that balance security and performance; second, optimize verification processes by adopting the "check first, compute later" model mentioned earlier to quickly reject invalid requests; third, implement tiered protection strategies, applying full verification processes only to critical operations while using simplified verification for non-sensitive operations. These

optimization strategies can control performance impact within acceptable ranges while maintaining core security properties.

Execution environment integrity is an inherent challenge for the dynamic token method. Since token verification processes typically occur within function instances, attackers theoretically could bypass verification mechanisms if they gain complete control of the execution environment. This limitation relates to the fundamental characteristics of serverless architecture: function instances are the smallest execution units with limited internal security boundaries. To address this challenge, a viable enhancement direction is to move core verification logic to independent trusted environments, such as dedicated security services or solutions based on trusted execution environments (TEEs). Although this approach may introduce additional architectural complexity and performance overhead, such trade-offs are usually reasonable for applications with high security requirements. Moreover, even with in-function verification, additional multi-factor authentication and behavioral analysis can significantly increase attack difficulty, forming effective defense-in-depth.

From development and operations perspectives, the dynamic token method does add certain implementation complexity. However, through appropriate tool support and abstraction encapsulation, adoption thresholds can be effectively lowered. Ideal implementations should provide developer-friendly SDKs and middleware, encapsulating complex token generation and verification logic into simple API calls. Integration with modern development frameworks and toolchains is also crucial; for example, integration with common serverless frameworks (such as serverless framework, AWS SAM, *etc.*,) can significantly simplify configuration and deployment processes. In terms of observability, integrating detailed logging and monitoring capabilities to enable development teams to effectively troubleshoot token-related issues is also a key factor in reducing operational complexity.

Analysis of applicable scenarios indicates that the dynamic token method is particularly suitable for the following cases: protecting sensitive operations in multi-tenant environments, applications processing high-value data, compliance scenarios requiring fine-grained audit capabilities, and financial and medical applications facing strict security reviews. Conversely, for applications with extreme low-latency requirements, simple functions without sensitive data access, or private deployment scenarios with already well-established internal security isolation, the value of the dynamic token method is relatively smaller, and implementation costs may exceed security benefits. Understanding these applicability boundaries helps organizations make informed security investment decisions.

Overall, the dynamic token encryption and decryption method demonstrates significant advantages in enhancing serverless application security, particularly excelling when addressing the various attack models analyzed previously. While its limitations objectively exist, most can be effectively mitigated through reasonable design choices and implementation strategies. Future research should focus on further reducing performance overhead, enhancing cross-platform compatibility, and integration with emerging security

technologies such as zero trust architecture, to drive serverless security towards higher levels.

## Application prospects and future research directions

The dynamic token encryption and decryption method demonstrates broad application prospects in serverless architecture and opens multiple valuable exploration paths for future research. This section explores the development potential and research directions of this method from dimensions such as application scenario expansion, technology integration, and security governance.

In terms of application scenarios, the dynamic token method is not only applicable to protecting system temporary keys but can also be extended to other sensitive information in environment variables, such as database credentials and API keys. As the serverless computing model becomes increasingly popular in modern application development, this method can seamlessly integrate with the increasingly rich and flexible security services provided by cloud service providers, offering comprehensive security protection for serverless applications. For industries with extremely high requirements for data security and privacy protection, such as finance and healthcare, the dynamic token method significantly enhances the granularity and flexibility of data encryption and access control by providing dynamically generated security tokens for each request, better adapting to the high security requirements of these industries.

Integration with emerging security architectures represents an important development direction. Zero Trust Architecture emphasizes the security concept of "never trust, always verify," which highly aligns with the design philosophy of dynamic tokens. Future research can explore using dynamic tokens as a key verification component in Zero Trust Architecture, implementing fine-grained authorization decisions based on request context. Specifically, dynamic tokens can integrate multi-dimensional trust signals including identity, device, network path, and behavior patterns to build a more comprehensive security decision framework. In edge computing scenarios, the main challenge for the dynamic token method is efficient implementation in resource-constrained environments. Future research needs to develop lighter-weight token generation and verification algorithms to adapt to the performance characteristics of edge nodes while maintaining necessary security strength.

In the field of security compliance and governance, the dynamic token method helps organizations meet increasingly stringent data protection regulatory requirements. For example, regulations such as the European Union's General Data Protection Regulation (GDPR) and the California Consumer Privacy Act (CCPA) require organizations to implement appropriate technical measures to protect personal data. Dynamic tokens can support these compliance requirements through their fine-grained access control capabilities and comprehensive audit logging. Particularly in terms of data access traceability and the principle of least necessary access, the dynamic token mechanism provides authorization control precise to the request level and detailed operation audit records, effectively supporting regulatory compliance. For organizations that need to comply with Cloud Security Alliance (CSA) STAR certification or financial industry PCI

DSS standards, future research can further explore how to map the dynamic token mechanism to specific control points of these standards, providing more direct compliance support.

Based on user survey feedback, we have identified several key improvement directions. First, simplifying the initial configuration experience is an important factor in increasing adoption rates. Future research can focus on developing more intelligent configuration assistants and template libraries to help developers quickly apply best practice configurations. Second, deep integration with mainstream development tools and frameworks is also crucial. By providing plugins and extensions for commonly used IDEs and CI/CD platforms, security controls can be seamlessly integrated into the development process, lowering adoption barriers. Additionally, the survey shows that users have a strong demand for clearer error messages and troubleshooting guides, which is also a focus for future work.

Progressive adoption strategy is an effective path for implementing complex security mechanisms. We recommend that organizations begin by identifying and protecting their most critical data assets, such as first applying to functions that process payment information, personal identity data, or business secrets. This gradual implementation approach not only distributes technical risk but also gives development teams time to learn and adapt. In terms of integration with existing security infrastructure, the dynamic token mechanism should be viewed as part of a defense-in-depth strategy rather than an isolated solution. Future research can explore collaborative working modes between dynamic tokens and security tools such as WAF, RASP, and cloud security configuration management to build a more complete serverless security protection system.

Security metrics research is the foundation for evaluating security enhancement effects. Existing security assessments often lack quantitative indicators specific to serverless security mechanisms. Future research should develop an evaluation framework applicable to the dynamic token method, including token strength indicators, token abuse resistance indicators, and security boundary measurement methods in different attack scenarios. This type of research not only helps objectively evaluate security improvement effects but also provides a more scientific basis for security investment decisions. Precise measurement standards are particularly important when considering the balance between security and performance.

In terms of performance optimization, future research can explore more efficient encryption algorithms, hardware acceleration mechanisms, and parallel computing frameworks, striving to minimize computational overhead and latency while ensuring security. Particularly noteworthy is the combination of hardware security modules (HSM) and trusted execution environments (TEE) with the dynamic token mechanism, which may provide higher levels of protection for sensitive operations while enhancing performance through hardware acceleration. Additionally, researching how to optimize token processing in cold start scenarios is also significant, with possible directions including preheating strategies, lazy loading mechanisms, and context caching technologies.

Regarding open source and ecosystem development, establishing open standards and reference implementations will help promote wider adoption. By open-sourcing core components, community wisdom can be gathered to improve algorithms and security while enhancing transparency and credibility. In terms of standardization, future work can promote the development of common specifications for serverless security tokens, ensuring interoperability between different implementations. Such standards should include aspects such as token format, lifecycle management, verification protocols, and security properties, providing a consistent security model for different platforms and vendors.

In conclusion, the dynamic token encryption and decryption method creates a promising research path for the field of serverless security. With the continued development of serverless architecture and the cross-integration of new technologies such as cloud computing, big data, artificial intelligence, and the Internet of Things, the dynamic token method will embrace broader development space and research hotspots. Through continuous innovation in performance optimization, intelligent adaptation, multi-scenario expansion, and cross-cloud collaboration, the dynamic token method is expected to become a key pillar in the serverless security protection system, providing users with comprehensive, intelligent, and flexible security services, helping the serverless computing model become a mainstream application architecture and solid foundation for digital transformation in the new era.

## CONCLUSION

This research deeply explores the significant contributions of dynamic token encryption and decryption methods in enhancing serverless application security from both theoretical and practical dimensions. Through systematic theoretical modeling, experimental validation, and practical application analysis, we propose an innovative security solution tailored to serverless architecture characteristics, providing new insights for addressing the adaptability challenges of static permission models in highly dynamic operating environments.

From a theoretical contribution perspective, this study establishes a theoretical framework for dynamic token security models, built around four core security attributes: time-effectiveness, context-binding, unpredictability, and non-reusability. Unlike traditional static authorization methods, this model incorporates function execution context and request-specific information into the security decision process, achieving precise permission control at the request level. This methodological innovation extends the perspective of serverless security research, filling research gaps in dynamic authorization mechanisms in existing literature. Particularly in connecting the theoretical model with the OWASP Serverless Top 10 security risk framework, this research establishes clear mapping relationships, demonstrating how dynamic token mechanisms specifically address major security threats in serverless environments.

In terms of practical contributions, this research demonstrates the feasibility and effectiveness of dynamic token mechanisms in real-world environments through prototype system implementation and multidimensional validation. Our experimental

evaluation covers various application patterns from simple functions to complex business scenarios, systematically quantifying the relationship between security enhancement and performance impact. Notably, as application complexity increases, the relative performance overhead introduced by dynamic token mechanisms gradually diminishes, becoming almost negligible in real business scenarios. This finding suggests that the method is not only theoretically sound but also practical for actual deployment, capable of maintaining good system performance while providing strong security guarantees.

The bidirectional interaction between theory and practice is a distinctive feature of this research. On one hand, our theoretical model provides clear guidance for prototype system implementation, with core designs such as token lifecycle management strategies and multi-factor verification mechanisms derived from the theoretical framework. On the other hand, findings from practical deployment have enriched the theoretical model, especially in terms of performance optimization and user experience, where feedback from actual applications has prompted us to adjust and refine our initial theoretical assumptions. This virtuous cycle between theory and practice makes the research outcomes more academically valuable and practically significant.

Despite the positive results achieved in this research, we honestly acknowledge its limitations. First, current experimental validation has been primarily conducted on specific cloud platforms, and research on cross-platform applicability requires further investigation. Second, although we have proposed possible solutions to the challenge of execution environment integrity, this fundamental issue has not been completely resolved, especially in extreme attack scenarios. Additionally, while the user experience survey provided valuable feedback, the sample composition was primarily small teams and individual developers, potentially not fully reflecting implementation challenges in large enterprise environments. These limitations suggest that while the dynamic token method is effective, it still needs to be validated and refined in broader scenarios.

From an industry practice perspective, this research provides practical guidance for enterprises adopting serverless technology. The research results indicate that dynamic token mechanisms can serve as an effective means to address serverless security pain points, particularly suitable for multi-tenant environments handling sensitive data. Our proposed progressive adoption strategy and integration methods with existing security infrastructure provide an actionable path for enterprise implementation, lowering the adoption threshold. Through case analyses in this research, we demonstrate application patterns of this method in different business scenarios, providing a reference framework for security teams to customize security solutions based on their specific needs.

In terms of academic contributions, this research provides new research perspectives and methodologies for the serverless security field. By applying dynamic authorization, context awareness, and defense-in-depth principles to serverless environments, we expand the boundaries of cloud security research. In particular, the approach of using function execution context as a security decision element provides an innovative direction for addressing authorization challenges in cloud-native environments. This research framework is not only applicable to serverless computing but also provides valuable

references for related fields such as microservice security and container security, with the potential to influence broader cloud-native security research.

Looking ahead, there remains vast space for research and application of dynamic token methods. As serverless architecture extends to more complex scenarios, security requirements will continue to evolve. We believe that subsequent research can continue to explore in several directions: first, deeply integrating dynamic token mechanisms with zero-trust architecture to build a more comprehensive cloud-native security framework; second, exploring token protection mechanisms based on hardware trusted roots to further enhance execution environment security; and third, developing formal verification techniques for dynamic token methods to provide more rigorous security guarantees. These research directions require interdisciplinary collaboration, combining knowledge from cryptography, system security, and software engineering to achieve breakthrough progress.

In conclusion, this research makes meaningful theoretical and practical contributions to the field of serverless security, proposing dynamic token encryption and decryption methods that provide new insights for addressing permission management challenges in serverless environments. We believe that as serverless technology continues to develop and find widespread application, this security enhancement method will play an increasingly important role, promoting serverless computing as a secure and reliable pillar of new-era digital infrastructure. Future research will continue to deepen and expand this field, driving the joint advancement of serverless security theory and practice.

### Funding
The authors received no funding for this work.

### Competing Interests
The authors declare that Sun Chenhao is employed by Alibaba Group. The other authors have no competing interests to declare.

### Author Contributions
- Yu Liu conceived and designed the experiments, performed the experiments, analyzed the data, performed the computation work, prepared figures and/or tables, authored or reviewed drafts of the article, and approved the final draft.
- Fu Li conceived and designed the experiments, performed the computation work, authored or reviewed drafts of the article, and approved the final draft.
- Chenhao Sun performed the experiments, analyzed the data, authored or reviewed drafts of the article, and approved the final draft.

### Data Availability
The dataset is available at figshare: Liu, Yu; Fu, Li; Chenhao, Sun (2025). Dataset for Dynamic Token Encryption: Design and Implementation. figshare. Dataset. https://doi.org/10.6084/m9.figshare.28931252.v1.

The code is available at GitHub and Zenodo:

- https://github.com/anycodes/DynamicTokenEncryption.

- Anycodes. (2025). anycodes/DynamicTokenEncryption: DynamicTokenEncryption V0.0.1 (0.0.1). Zenodo. https://doi.org/10.5281/zenodo.15726188.

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
