# Peer review of "Dynamic token encryption for preventing permission leakage in serverless architectures"

_PeerJ Computer Science, doi:10.7717/peerj-cs.3029_

## Round 0.1 · original submission · Major Revisions

Please consider all required reviewer comments to increase the quality of your paper.

·

Basic reporting

The paper highlights a critical gap in serverless security, focusing on the risks of static permission models in highly dynamic runtime environments. It emphasizes how models based on static execution roles struggle to adapt to changing conditions, posing security risks.

Certain technical descriptions, such as "call chain features" and "behavior pattern verification," are introduced without sufficient explanation.

Figure 1 (Dynamic Token Encryption Architecture) is difficult to follow. The flow of the dynamic process is unclear. It would be better to include steps numbering.

The introduction is weak due to a lack of citations supporting key claims. Similarly, the related work section lacks sufficient references to substantiate its statements, reducing the rigor of the literature review.

No replication code, configuration files, or survey raw data are provided, hindering independent verification of the results.

It is hard to follow the key findings of the paper, try to make it clear in boxes after each experiment/finding.

Experimental design

I appreciate the validation of the proposed idea through a survey; however, the methodology lacks transparency. The sampling criteria, such as developer expertise, are not clearly defined. While the survey questions are provided in Table 5, the absence of raw response data and analysis scripts limits the reproducibility and validity of the findings.

The cold/hot start performance tests (Tables 1–4) focus on latency and memory overhead but lack justification for their relevance to the paper’s security claims. The connection between startup time and security efficacy is not explained. Looks not relevent to the paper focus !

Validity of the findings

The study makes a meaningful contribution to serverless security by addressing permission leakage with dynamic tokens.

Only Theoretical Justifications for Security Strength. The Qualitative and Quantitative Evaluation of the Security Analysis section is purely theoretical, lacking empirical validation through real-world attack simulations or penetration testing. While the paper identifies potential security threats and discusses how dynamic tokens mitigate them, it does not provide concrete experimental evidence, structured threat modeling, or quantitative security metrics to measure effectiveness.


User Survey Limitations:
With only 111 participants and no information on their selection criteria, the survey results may not be representative of the broader serverless development community.

Additional comments

The lack of citations for key claims in the introduction and related work weakens the credibility of the argument.

The experimental design needs better justification, particularly for the cold/hot start performance tests, which do not directly relate to security.

I suggest using a public dataset of serverless functions to empirically test the proposed security solution.

No replication code is provided, limiting the ability for others to validate the results.

Reviewer 3 ·

Basic reporting

The study offers a approach for improving serverless security by using dynamic token encryption and decryption. To provide a more reliable basis for the suggested approach, the literature review has to be improved. It is noticed that there are improper citations to references to current authentication methods.

Experimental design

no comment

Validity of the findings

The duration of token validity prior to expiration should be explicitly specified in the article, along with an analysis of the decisions between system overhead and security.
To validate the security claims, an analysis of the system’s resilience against common threats (e.g., token replay attacks, leakage, man-in-the-middle attacks, and privilege escalation) to be included. Although the idea of separating token verification from function execution is intriguing, further information regarding its use and efficacy is required.

Additional comments

The manuscript presents a promising method for enhancing serverless security via dynamic token encryption and decryption is presented. By adding a temporary, request-specific security token, the suggested paradigm successfully improves fine-grained access control and data protection, which fits in nicely with serverless computing's elastic nature.

Reviewer 4 ·

Basic reporting

In this study, the authors propose a dynamic token-based access control mechanism. This mechanism addresses the security risks of static authorization models in serverless architectures. The proposed approach manages token lifetimes and contextual data dynamically. It aims to enhance security through a variety of supporting techniques. Various performance metrics of the application are being tested across different platforms.

• Some explanations in the introduction section are not sufficiently detailed. The reasons why existing solutions are inadequate should be elaborated further. Moreover, while data leakage incidents are mentioned, no concrete examples are provided to support these claims.
• Some terms are used inconsistently throughout the study (e.g., "dynamic token mechanism" vs. "dynamic token encryption-decryption method"). A consistent terminology should be adopted by selecting a standardized form for such terms.
• The figures and tables presented in the study lack sufficient clarity. More explicit explanations should be provided within the text to support their interpretation. In particular, complex visuals such as Figure 1 should be revised or simplified to enhance readability and overall comprehension.
• Some of the information and statistical data presented in the article are not supported by adequate citations, which may reduce their perceived credibility. To strengthen the validity of these claims, they should be backed by well-established and relevant references.
• Certain sections of the paper contain repetitive content. To improve the overall coherence and readability, these redundant parts should be reviewed and refined to achieve a more concise and effective presentation.
• The proposed "dynamic token" approach lacks a clear explanation of how it differs from existing mechanisms such as JWT (JSON Web Tokens) and HMAC-based one-time tokens. The novelty and distinct features of the proposed system should be explicitly articulated in comparison with these well-established methods.
• The proposed "dynamic token" approach does not clearly explain how it differs from existing methods such as JWT (JSON Web Tokens) and HMAC-based one-time tokens. The distinctions between the proposed system and these widely used mechanisms should be clarified. Additionally, the innovative aspects of the approach need to be explicitly stated.

Experimental design

• The study states that nonces are used as part of the anti-replay mechanism. However, it does not provide sufficient detail on how the nonces are handled, how frequently they are reset, or how the system manages this process. The management strategy and validity period of the nonces should be clearly defined.
• The presented survey results do not provide sufficient information on how the study was conducted. It is unclear how participants were selected, as well as their basic demographic details such as age, profession, and the industries they represent.
• The encryption methods used for token generation and the associated technical details (e.g., key length, hash algorithm) are not sufficiently explained. Although different sections of the study mention the need for various algorithms, it is not explicitly stated which algorithms were chosen and how they were implemented.
• The approach of conducting a limited number of tests per scenario and selecting only the highest and lowest values raises concerns about statistical validity. To ensure the reliability of the results, a more comprehensive statistical analysis should be performed, potentially by increasing the sample size and incorporating a broader range of values.
• The claimed novelty of the study primarily consists of the combination of common security principles (timestamp, nonce, hash). Whether this constitutes a sufficient innovation should be discussed in greater detail.
• The study mentions the use of hash functions such as MD5 and SHA-1. Although these functions may offer performance benefits, they are now considered cryptographically insecure. The security implications of these choices should be thoroughly examined.

Validity of the findings

• The study claims that dynamic tokens are more secure than static authorization mechanisms. However, additional comparative analyses should be included to support this claim.
• While the study theoretically explains the security advantages of dynamic tokens, the security model itself is not sufficiently detailed.
• The lack of comparison with traditional access control methods makes it difficult to determine the strengths and weaknesses of the proposed approach.

Reviewer 5 ·

Basic reporting

This research work is coherent and has a strong and relevant technical contribution. It proposes a solution to the problem that has been a challenge in serverless computing. The research finds the possibility of key leakage because of the static permission models, and presents a new approach for dynamic token generation and verification. The paper is well organized, written in professional English (with some LaTeX encoding issues that need fixing), and presents a good background and literature review.

The methodology is original and rigorous. It captures system architecture, encryption processes, and verification processes outlining enough detail for replication. The experimental design is comprehensive and includes security modeling, case studies, and performance evaluation across Alibaba and Tencent Cloud, which add relevance to the work.

The authors extended the applicability of the paper by including a user study in which 111 serverless developers were surveyed. The performance results are favorable; however, the discussion concerning increased cold start times, especially on the Alibaba Cloud, from a user experience perspective is needed. The authors also note that execution environment integrity is a limitation, but more explanation on what mitigations could be used is necessary.

Experimental design

The implementing framework provided in this research offers a systematic methodology for tackling the issue of permission leakage on serverless architectures, although it has a few methodological and technical drawbacks. The work which aims to verify the problem’s relevance suffers from a lack of transparency concerning sample size, population, and sampling methodology which limits its generalizability. Similarly, the prototype is only described as having been implemented on “mainstream serverless platforms” without identifying which ones, or how they were integrated, which limits evaluation of potential real-world impact. The lack of clearly defined attack vectors, quantitative, detection rates, and relevant comparisons to security standards weakens the validation’s credibility. Claims made regarding performance are not supported by definitive figures on latency, resource utilization, or benchmarks against other proposed solutions like AWS KMS and Leakless.

In addition, no open-source code or data is made available, nor is provided the documentation required for reproducibility, a complete description of the system’s architecture, including the token generation algorithm used, encryption schema, and workload parameters—is absent. The greatest contribution of the work is arguably this novel combination of call chain analysis and behavioral mapping for multi-factor token verification. However, fully substantiate its claims, the research requires greater methodological transparency, technical specificity, and comparative benchmarking. Addressing these gaps would significantly enhance the research’s impact and credibility.

Validity of the findings

The article highlights an important security issue with serverless computing: the potential for permission and key leakage because of static and overly permissive execution roles. To address this, the authors suggest a dynamic access control method based on token refinement that produces highly complex, single-use tokens at runtime. These tokens grant access control logic and user security policies to enable fine-grained, posture-aware dynamic enforcement and request-level identity verification. Moreover, the scheme includes a multi-factor token validation method that uses call chain feature analysis and behavior pattern monitoring to protect against multiple security threats.

This work features a novel approach of merging dynamic token creation with behavioral checks for better access control in serverless environments. The authors claim to have built a prototype on popular serverless platforms and performed extensive testing alongside social surveys, qualitative analyses, and claims to have realized better security without significant performance costs.

Nevertheless, the lack of detailed methodology paired with inadequate quantitative assessment of results undermines the work’s contribution. It is unclear how the readers would understand the work without an explanation for the structuring of the implementation, choice of platform, and alternative benchmarks which are all critical to achieving reliable results. Moreover, comparisons with existing solutions and benchmarks are absent, reducing the ability to fully assess the work's advantages.

Overall, the paper contributes valuable ideas to serverless security research but requires more thorough experimental validation and transparency to strengthen its practical relevance and credibility.

Additional comments

This paper addresses a critical and challenging issue in serverless computing, the risk of permission and key leakage due to static, overly permissive access models. It proposes a dynamic, context-aware access control framework that leverages runtime token generation, call chain analysis, and behavioral monitoring to enable fine-grained, posture-aware security enforcement. The approach is innovative, blending technical depth with practical relevance, and is supported by architectural modeling, encryption flow design, and empirical testing on major cloud platforms like Alibaba and Tencent Cloud. A user study involving serverless developers further strengthens the contextual grounding of the research.

However, the paper's potential is weakened by gaps in transparency and technical depth. Critical details such as the specific platforms used, integration strategies, token generation algorithms, encryption schemas, and workload parameters are omitted. While performance improvements are claimed, the absence of quantitative benchmarks, latency metrics, and comparisons with existing solutions like AWS KMS and Leakless limits the credibility of those claims. Moreover, the lack of publicly available code or implementation details leads to vague understanding about the work. The discussion of limitations, particularly around execution environment integrity and cold start impacts, remains underdeveloped.

Despite these shortcomings, the paper introduces a compelling direction for enhancing serverless security. With improved methodological rigor and clearer validation, it could make a significant impact in both academic and practical domains.

---

## Round 0.2 · Minor Revisions

Please update your manuscript based on the reviewer's comments

·

Basic reporting

While the manuscript has been substantially improved and largely meets publication standards, I suggest the following minor revisions be considered:
- Include numbering of the steps inside the figure 1, same as you did inside the text.
- Figure 4, it appears as a screenshot.
- Figure 5: Replace the questions number with short descriptive.

Several references are not well organized.

- Incorrect Citation [18], The correct reference is:

Calles, Miguel A. Serverless Security: Understand, Assess, and Implement Secure and Reliable Applications in AWS, Microsoft Azure, and Google Cloud. Apress, 2020.

- Duplicate Citations [12] and [13]
These two citations refer to the same source and should be merged into one.

- The Performance Optimization Section (3.3)
Summarize the specific optimizations, including the use of lightweight hash functions, tiered verification logic, and others.
I suggest you present them in a table format that shows, original approach vs. optimized version. You can add also the purpose or motivation of each optimization.

Experimental design

no comment

Validity of the findings

no comment

Reviewer 3 ·

Basic reporting

1.references are not cited properly
2. Lack of literature review

Experimental design

-

Validity of the findings

The author didnot addressed the suggestions that were previously provided.

Additional comments

-

Reviewer 5 ·

Basic reporting

This research work is coherent and has a strong and relevant technical contribution. It proposes a solution to the problem that has been a challenge in serverless computing. The research finds the possibility of key leakage because of the static permission models, and presents a new approach for dynamic token generation and verification. The paper is well organized, written in professional English (with some LaTeX encoding issues that need fixing), and presents a good background and literature review.

The methodology is original and rigorous. It captures system architecture, encryption processes, and verification processes outlining enough detail for replication. The experimental design is comprehensive and includes security modeling, case studies, and performance evaluation across Alibaba and Tencent Cloud, which add relevance to the work.

The authors extended the applicability of the paper by including a user study in which 111 serverless developers were surveyed. The performance results are favorable; however, the discussion concerning increased cold start times, especially on the Alibaba Cloud, from a user experience perspective is needed. The authors also note that execution environment integrity is a limitation, but more explanation on what mitigations could be used is necessary.

Experimental design

As per the review comments the authors have updated the article.
A detailed description of the system architecture, including the token generation algorithm, encryption schema, and workload parameters, has now been added. Furthermore, authors have expanded the methodological discussion to improve transparency and included comparative benchmarking results to substantiate our claims. I believe these enhancements significantly strengthen the clarity, impact, and technical rigor of our work.

Thus, the changes found to be satisfactory.

Validity of the findings

As per the review comments the authors have updated the article.
The authors have significantly expanded the methodology section to include a detailed explanation of the implementation structure, choice of platform, and rationale behind key design decisions. We have also incorporated comparative analysis with existing solutions and benchmarks to quantitatively assess our results. These additions aim to enhance the reliability, clarity, and evaluative strength of the work, addressing the reviewer’s concerns comprehensively.

Thus, the changes found to be satisfactory.

Additional comments

No further revision needed.

---

## Round 0.3 · accepted · Accept

Since all required comments are accepted from reviews, I would like to inform you that your paper is ready for publication. Whenever you finalize your paper, ensure that there are no syntactical or grammatical mistakes.

·

Basic reporting

Authors addressed all my comments. It would be great if they replace all current Figures with PDF version for readability and manuscript quality.

Experimental design

no comment

Validity of the findings

no comment

Reviewer 3 ·

Basic reporting

The revised manuscript is well structured. Still references are not cited properly.

Experimental design

-

Validity of the findings

The Updated manuscript has enhanced significantly in establishing the validity of its findings.The results have been presented in a more credible and substantiated way overall.

Additional comments

The authors have successfully refined the manuscript, addressing key technical and structural concerns raised in the previous review.